# LEARNING NON-LINEAR TRANSFORM WITH DISCRIMINATIVE AND MINIMUM INFORMATION LOSS PRIORS

## ABSTRACT

This paper proposes a novel approach for learning discriminative and sparse representations. It consists of utilizing two different models. A predefined number of non-linear transform models are used in the learning stage, and one sparsifying transform model is used at test time. The non-linear transform models have discriminative and minimum information loss priors. A novel measure related to the discriminative prior is proposed and defined on the support intersection for the transform representations. The minimum information loss prior is expressed as a constraint on the conditioning and the expected coherence of the transform matrix. An equivalence between the non-linear models and the sparsifying model is shown only when the measure that is used to define the discriminative prior goes to zero. An approximation of the measure used in the discriminative prior is addressed, connecting it to a similarity concentration. To quantify the discriminative properties of the transform representation, we introduce another measure and present its bounds. Reflecting the discriminative quality of the transform representation we name it as discrimination power.

To support and validate the theoretical analysis a practical learning algorithm is presented. We evaluate the advantages and the potential of the proposed algorithm by a computer simulation. A favorable performance is shown considering the execution time, the quality of the representation, measured by the discrimination power and the recognition accuracy in comparison with the state-of-the-art methods of the same category.

## 1 INTRODUCTION

Learning a transform that provides sparse and discriminative representation is an active domain of research in various areas, some of which are data processing, pattern recognition, image processing, language modeling, text analysis and gene separation. A class of algorithms proposed by Kreutz-Delgado et al. (2003); Mairal et al. (2009); Bengio et al. (2012); Gangeh et al. (2015); Mairal et al. (2008); Jiang et al. (2011); Guo et al. (2012); Cai et al. (2014) and Liu et al. (2016) for learning discriminative sparse representations have been shown to perform well across various learning tasks. A subclass of them known as *discriminative dictionary learning* (DDL) Guo et al. (2012); Jiang et al. (2013); Cai et al. (2014); Shekhar et al. (2014); Xu et al. (2015); Liu et al. (2016); Bengio et al. (2012); Gangeh et al. (2015); Jiang et al. (2016) and Vu & Monga (2016) addresses the estimate of the dictionary in a supervised manner such that the representation w.r.t. words (vectors) from the resulting dictionary (vector set) is discriminative.

Most of the DDL methods *synthesize* the data sample $k$ from class $c$, *i.e*, $\mathbf{x}_{c,k} \in \Re^N$ as an approximation by a linear combination $\mathbf{y}_{c,k} \in \Re^M$ (referred to as a sparse data representation $\|\mathbf{y}_{c,k}\|_0 << M$) of a few words (vectors $\mathbf{d}_m$), from a dictionary (vector set) $\mathbf{D} \in \Re^{N \times M}$, *i.e.*, $\mathbf{x}_{c,k} = \mathbf{D}\mathbf{y}_{c,k} + \mathbf{v}_{c,k}$, $\mathbf{v}_{c,k} \in \Re^N$, with $\mathbf{v}_{c,k}$ denoting the approximation error. It is important to highlight that with the *synthesis model* approach the *data reconstruction* is addressed.

The differences between the DDL methods Guo et al. (2012); Jiang et al. (2013); Cai et al. (2014); Shekhar et al. (2014); Gangeh et al. (2015); Xu et al. (2015); Liu et al. (2016); Bengio et al. (2012); Jiang et al. (2016) and Vu & Monga (2016) are determined by the prior defined on the sparse representation and the prior defined for the relations between the sparse representations for the data samples from the same/different classes. The discrimination is enforced by replacing the prior with

a structural constraint on the dictionary or imposing a discriminative term on the sparse representations. Additionally, some works by Mairal et al. (2008); Guo et al. (2012) and Taalimi et al. (2015) consider even a joint estimation/learning of a dictionary, sparse representation, and classifier by using iterative alternating minimization strategy. The manuscripts by Bengio et al. (2012); Cai et al. (2014) and Gangeh et al. (2015) give comprehensive overview covering different approaches.

## 1.1 OPEN ISSUES

The general open issue for DDL methods is the computational complexity w.r.t. the optimal dictionary/transform learning and the discriminative encoding, since the sparse representation in the *synthesis model* is a solution to an *inverse problem*.

An additional open issue with most of the proposed approaches Guo et al. (2012); Jiang et al. (2013); Cai et al. (2014); Gangeh et al. (2015); Liu et al. (2016), Bengio et al. (2012); Jiang et al. (2016); Vu & Monga (2016) is that there is no formal notion to measure the discriminative properties. Therefore, there are no means that provide a quantitative evaluation of the quality of the representation, other than the performance of a classifier used on top of the representation.

Concerning the specifics in the discriminative constraints, Yang et al. (2011b) proposed a synthesis model with a discriminative fidelity term and Fisher discriminant constraints, where the within-class scatter and the between-class scatter of the representation is minimized and maximized, respectively. The authors Vu & Monga (2016) proposed an extension considering a low-rank constraint on the dictionary. An approach by Guo et al. (2013) used a synthesis model with a constraint on the pair-wise relation between the sparse representation expressed by $\ell_2$ distance metric. The methods reported by Yang et al. (2011b) Vu & Monga (2016) and Guo et al. (2013) take into account assumption on the metric by defining the scatter and the pair-wise relations. Therefore, they constrain the space of the representation, which essentially is determined by the dictionary. However, these works do not consider whether the used metric is optimal w.r.t. the sparse representation.

The method proposed by Liu et al. (2016) finds a dictionary under which the representation of a data sample from the same class $c$ have a common sparse structure by minimizing the size of the support overlap for the representation from different classes. Assuming $\mathbf{y}_{c1,k1} \in \Re^M$ and $\mathbf{y}_{c2,k2} \in \Re^M$ are two sparse representations for two data samples $\mathbf{x}_{c1,k1} \in \Re^N$ and $\mathbf{x}_{c2,k2} \in \Re^N$, from two classes $c1$ and $c2$, they proposed a similarity measure defined by empirical expectation on $\|\mathbf{y}_{c1,k1} \odot \mathbf{y}_{c2,k2}\|_0$, where $\odot$ represents the Hadamar product. Note that two transform data samples $\mathbf{y}_{c1,k2}$ and $\mathbf{y}_{c2,k2}$ that have small support overlap $\|\mathbf{y}_{c1,k1} \odot \mathbf{y}_{c2,k2}\|_0 = s, s << M$, might not necessarily be similar or dissimilar, *i.e.*, $\mathbf{y}_{c1,k1} = \mathbf{y}_{c2,k2}$ and $\mathbf{y}_{c1,k1} = -\mathbf{y}_{c2,k2}$ with $\|\mathbf{y}_{c1,k1}\|_0 = \|\mathbf{y}_{c2,k2}\|_0 = s$ and $s$ small.

## 1.2 APPROACH AND MOTIVATIONS

**Model** Instead of addressing a synthesis model where the data reconstruction is targeted and the estimation of the discriminative representation is an *inverse problem*[1] we present a novel, alternative approach. That is we propose *non-linear transform models* in the learning stage and a *sparsifying transform model* Rubinstein et al. (2010), Rubinstein et al. (2013), Rubinstein & Elad (2014) and Ravishankar & Bresler (2014) for testing. The sparsifying transform model assumes that the data sample $\mathbf{x}_{c,k}$ is approximately sparsifiable under a linear transform $\mathbf{A} \in \Re^{M \times N}$, *i.e.*, $\mathbf{A}\mathbf{x}_{c,k} = \mathbf{y}_{c,k} + \mathbf{z}_{c,k}, \mathbf{z}_{c,k} \in \Re^M$, where $\mathbf{y}_{c,k}$ is sparse $\|\mathbf{y}_{c,k}\|_0 << M$. It also represents a generalization of the analysis model Rubinstein et al. (2010; 2013); Ravishankar & Bresler (2014); Rubinstein & Elad (2014).

The proposed non-linear transform model is an extension to the sparsifying transform model that considers additional assumptions. Both of the models used in this paper address a *direct problem*, where the estimation of the discriminative representation represents a low complexity constrained projection problem. Additionally, since these models have no restrictions on the transform representation to be in the column space of the dictionary (transform matrix $\mathbf{A}$), they allow more freedom in modeling and imposing constraints on the transform representation [2].

---

[1] Note that a solution to an inverse problem has a high computational complexity if the dimensionality of the dictionary (transform matrix) or the data dimensionality is high.

[2] In fact it allows modeling other non-linearity also, *i.e.*, ReLu can be modeled as a transform representation.

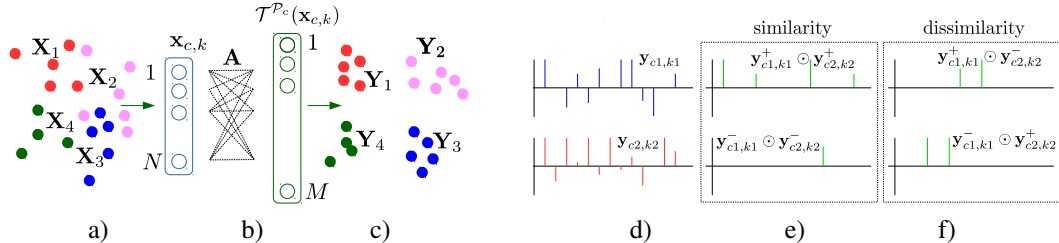

Figure 1: a) Data samples $\mathbf{X}_c, c \in \{1, 2, 3, 4\}$ from four different classes, b) given a $k$-th data sample $\mathbf{x}_{c,k}$ from class $c$, the non-linear transform is represented as two step operation: linear mapping $\mathbf{A}\mathbf{x}_{c,k}$ (*step 1*) followed by an element wise thresholding function $\mathbf{y}_{c,k} = \mathcal{T}^{\mathcal{P}_c}(\mathbf{A}\mathbf{x}_{c,k})$ (*step 2*), c) the transform data samples $\mathbf{Y}_c, c \in \{1, 2, 3, 4\}$, d) two transform representations $\mathbf{y}_{c1,k1}$ and $\mathbf{y}_{c2,k2}$, e) the resulting Hadamar products $\mathbf{y}^+_{c1,k1} \odot \mathbf{y}^+_{c2,k2}$ and $\mathbf{y}^-_{c1,k1} \odot \mathbf{y}^-_{c2,k2}$ on the support intersection for the similarity contribution and f) the resulting Hadamar products $\mathbf{y}^+_{c1,k1} \odot \mathbf{y}^-_{c2,k2}$ and $\mathbf{y}^-_{c1,k1} \odot \mathbf{y}^+_{c2,k2}$ on the support intersection for the dissimilarity contribution between $\mathbf{y}_{c1,k1}$ and $\mathbf{y}_{c2,k2}$.

**Prior and its measure** In the learning stage, central in the non-linear transform models is the novel parametric measure for a discriminative prior. It is defined on the *support intersection* for the transform representations. The first motivation behind the used measure is that the support intersection of the transform data allows more freedom in imposing regularization on the discriminative properties without taking into account any additional assumptions. Second, by approximating the parametric mesure with an non-parametric one the focus of the regularization is directly put on the contributing components for similarity/dissimilarity. Consider the measure $(\mathbf{y}^+_{c1,k1})^T \mathbf{y}^+_{c2,k2} + (\mathbf{y}^-_{c1,k1})^T \mathbf{y}^-_{c2,k2}$ between two transform representations $\mathbf{y}_{c1,k1}$ and $\mathbf{y}_{c2,k2}$ where $\mathbf{y}_{c1,k1} = \mathbf{y}^+_{c1,k1} - \mathbf{y}^-_{c1,k1}$ where $\mathbf{y}^+_{c1,k1} = \max(\mathbf{y}_{c1,k1}, \mathbf{0})$ and $\mathbf{y}^-_{c1,k1} = \max(-\mathbf{y}_{c1,k1}, \mathbf{0})$. Note that $(\mathbf{y}^+_{c1,k1})^T \mathbf{y}^+_{c2,k2} + (\mathbf{y}^-_{c1,k1})^T \mathbf{y}^-_{c2,k2} = \|\mathbf{y}^+_{c1,k1} \odot \mathbf{y}^+_{c2,k2}\|_1 + \|\mathbf{y}^-_{c1,k1} \odot \mathbf{y}^-_{c2,k2}\|_1$ captures the only contribution for the similarity (whereas $\|\mathbf{y}^+_{c1,k1} \odot \mathbf{y}^-_{c2,k2}\|_1 + \|\mathbf{y}^-_{c1,k1} \odot \mathbf{y}^+_{c2,k2}\|_1$ captures the only contribution for the dissimilarity) between the vectors $\mathbf{y}_{c1,k1}$ and $\mathbf{y}_{c2,k2}$. Moreover, $\mathbf{y}^T_{c1,k1}\mathbf{y}_{c2,k2} = \|\mathbf{y}_{c1,k1} \odot \mathbf{y}_{c2,k2}\|_1$, only if the dissimilarity contribution $-(\mathbf{y}^-_{c1,k1})^T \mathbf{y}^+_{c2,k2} - (\mathbf{y}^+_{c1,k1})^T \mathbf{y}^-_{c2,k2}$ for the vectors $\mathbf{y}_{c1,k1}$ and $\mathbf{y}_{c2,k2}$ is 0. That is $\mathbf{y}^T_{c1,k1}\mathbf{y}_{c2,k2} = (\mathbf{y}^+_{c1,k1})^T \mathbf{y}^+_{c2,k2} + (\mathbf{y}^-_{c1,k1})^T \mathbf{y}^-_{c2,k2} - (\mathbf{y}^-_{c1,k1})^T \mathbf{y}^+_{c2,k2} - (\mathbf{y}^+_{c1,k1})^T \mathbf{y}^-_{c2,k2} \leq (\mathbf{y}^+_{c1,k1})^T \mathbf{y}^+_{c2,k2} + (\mathbf{y}^-_{c1,k1})^T \mathbf{y}^-_{c2,k2}$, now if $-(\mathbf{y}^-_{c1,k1})^T \mathbf{y}^+_{c2,k2} - (\mathbf{y}^+_{c1,k1})^T \mathbf{y}^-_{c2,k2}$ is zero then $\|\mathbf{y}_{c1,k1} \odot \mathbf{y}_{c2,k2}\|_1 = (\mathbf{y}^+_{c1,k1})^T \mathbf{y}^+_{c2,k2} + (\mathbf{y}^-_{c1,k1})^T \mathbf{y}^-_{c2,k2} = \mathbf{y}^T_{c1,k1}\mathbf{y}_{c2,k2}$. Third, the expectation $\mathbb{E}[\|\mathbf{y}^-_{c1,k1} \odot \mathbf{y}^-_{c2,k2}\|_1 + \|\mathbf{y}^+_{c1,k1} \odot \mathbf{y}^+_{c2,k2}\|_1]$ captures the *concentration of similarity*. Therefore, it provides the possibility to define a formal notion that quantifies the discriminative properties. Fourth, $(\mathbf{y}^+_{c1,k1})^T \mathbf{y}^+_{c2,k2} + (\mathbf{y}^-_{c1,k1})^T \mathbf{y}^-_{c2,k2}$ is not ambiguous w.r.t. a notion for similarity/dissimilarity between two sparse representations $\mathbf{y}_{c1,k1}$ and $\mathbf{y}_{c2,k2}$. This is because the support intersections for the positive and the negative components $(\mathbf{y}^+_{c1,k1})^T \mathbf{y}^+_{c2,k2} = \|(\mathbf{y}^+_{c1,k1}) \odot \mathbf{y}^+_{c2,k2}\|_1$ and $(\mathbf{y}^-_{c1,k1})^T \mathbf{y}^-_{c2,k2} = \|(\mathbf{y}^+_{c1,k1})^T \mathbf{y}^-_{c2,k2}\|_1$, respectively, are considered separately. In addition, taken into account is the strength on the support intersection, defined as $\|\mathbf{y}_{c1,k1} \odot \mathbf{y}_{c2,k2}\|_2^2$. Its empirical expectation $\sum_{c,k} \|\mathbf{y}_{c1,k1} \odot \mathbf{y}_{c2,k2}\|_2^2 \sim \mathbb{E}[\|\mathbf{y}_{c1,k1} \odot \mathbf{y}_{c2,k2}\|_2^2]$ captures the expected strength on the support intersection for that set. A schematic diagram of the transform and the main idea behind the proposed concept are shown in Figure 1, $a)$, $b)$ and $c)$. On Figure 1 $d)$, $e)$ and $f)$ are given illustrative examples for the support intersections between the positive and negative component of two vectors $\mathbf{y}_{c1,k1}$ and $\mathbf{y}_{c2,k2}$ in the transform domain.

A learning algorithm is presented using the proposed model with discriminative and minimum information loss priors. To quantify the discriminative properties, we introduce a measure named as *discrimination power*, which reflects the discriminate properties of the representation for a dataset. In addition, we present its lower and upper bounds, which depend on the parameters of the transform. On the practical side, the advantages and the potential of the proposed algorithm are demonstrated by a numerical experiments using the Extended YALE B Georghiades et al. (2001), AR Martínez & Benavente (1998), Norb LeCun et al. (2004), Coil-20 Nene et al. (1996), Clatech101 LeCun et al. (2008), UKB Nistér & Stewénius (2006) and MNIST Lecun & Cortes datasets.

## 1.3 NOTATIONS

A scalar variable is denoted using $x$, a vector is denoted by a bold, low caps symbols $\mathbf{x}$, a matrix by bold, upper cap symbol $\mathbf{A}$. A single element from a vector (or matrix) is denoted as $x(n)$ (or $A(m, n)$). A set is denoted by a calligraphic symbol, *i.e*, $\mathcal{S}$. The $\ell_p-$norm is denoted as $\|.\|_p$ and the nuclear norm as $\|.\|_*$. The $\odot$ symbol represents the Hadamard product. Throughout the paper it is assumed that a set of data samples $\mathbf{X} = [\mathbf{X}_1, \mathbf{X}_2, ..., \mathbf{X}_C] \in \Re^{N \times L}, L = CK$ from $C$ classes is given and that every class $c \in \mathcal{C} = \{1, 2, 3, ..., C\}$ has $K$ samples, $\mathbf{X}_c = [\mathbf{x}_{c,1}, \mathbf{x}_{c,2}, ..., \mathbf{x}_{c,K}] \in \Re^{N \times K}$, $\mathbf{x}_{c,k} \in \Re^N$, $\forall c \in \mathcal{C}, \forall k \in \mathcal{K} = \{1, 2, ..., K\}$. We denote the transform data as $\mathbf{Y} = [\mathbf{Y}_1, \mathbf{Y}_2, ..., \mathbf{Y}_C] \in \Re^{M \times L}$, where $\mathbf{Y}_c = [\mathbf{y}_{c,1}, \mathbf{y}_{c,2}, ..., \mathbf{y}_{c,K}] \in \Re^{M \times K}$ and $\mathbf{y}_{c,k} \in \Re^M$. We denote $\mathbf{X}_{\backslash \{k \in c\}} = [\mathbf{X}_1, \mathbf{X}_2, ..\mathbf{X}_{c,\backslash k}, ...\mathbf{X}_C] \in \Re^{N \times (L-1)}$ as the matrix that has all the columns of $\mathbf{X}$, except the column $\mathbf{x}_{c,k} \in \Re^N$, where $\mathbf{X}_{c,\backslash k} = [\mathbf{x}_{c,1}, \mathbf{x}_{c,2}, ..., \mathbf{x}_{c,k-1}, \mathbf{x}_{c,k+1}, ..., \mathbf{x}_{c,K}] \in \Re^{N \times (K-1)}$ is a matrix that has all the columns of block $\mathbf{X}_c$, except the column $\mathbf{x}_{c,k}$, $\forall c \in \mathcal{C}$ and $\forall k \in \mathcal{K}$. We let $\mathcal{N} = \{1, 2, ..., N\}$ and $\mathcal{M} = \{1, 2, ..., M\}$.

## 2 LEARNING NON-LINEAR TRANSFORM WITH DISCRIMINATIVE AND MINIMUM INFORMATION LOSS PRIORS

The proposed approach has two operational modes: learning and test. It considers two different models. A predefined number of non-linear transforms are used in the learning mode and one sparsifying transform is used for test time.

### 2.1 THE PARAMETRIC NON-LINEAR TRANSFORM MODELING

Considering the learning mode, we assume that for every class $c \in \mathcal{C}$ there exist one non-linear transform defined by a set of parameters $\mathcal{P}_c = \{\mathbf{A} \in \Re^{M \times N}, \boldsymbol{\tau}_c \in \Re^M\}, \forall c \in \mathcal{C}[3]$. All nonlinear transforms described by $\{\mathcal{P}_1, ..., \mathcal{P}_C\}$ share the linear map $\mathbf{A}$ and have distinct parameters $\boldsymbol{\tau}_c$. One $\mathcal{P}_c$ with $\boldsymbol{\tau}_c$ is related to only one class $c$. It is assumed that the parameters $\boldsymbol{\tau}_c$ are spread far apart in the transform domain. In addition, when all the non-linear transforms are applied to the corresponding class samples then the transform data samples are separable w.r.t. the different classes, in the transform domain.

As far as the non-linear transform we focus on transforms expressible by a two-steps operation, consisting of a linear mapping (step 1) followed by an element-wise non-linearity (step 2):

$$\mathbf{x}_{c,k} \xrightarrow[\text{step 1}]{\mathbf{A}} \mathbf{A}\mathbf{x}_{c,k} \xrightarrow[\text{step 2}]{\mathcal{H}_{\boldsymbol{\tau}_c}(.)} \mathbf{y}_{c,k}, \tag{1}$$

where $\forall \mathbf{w}_{c,k} = \mathbf{A}\mathbf{x}_{c,k} \in \Re^M, \mathcal{H}_{\boldsymbol{\tau}_c}(\mathbf{w}_{c,k}) = \text{sign}(\mathbf{w}_{c,k}) \odot \max(|\mathbf{w}_{c,k}| - \boldsymbol{\tau}_c, \mathbf{0}) : \Re^M \to \Re^M$, represents a non-linear thresholding function with parameters $\boldsymbol{\tau}_c \in \Re^M$. We also have to mention that the thresholding is done with different thresholding parameters $\tau_c(m)$ for the corresponding different transform dimensions $m \in \mathcal{M}$.

At testing time we use one, common sparsifying transform defined by a set of parameters $\mathcal{P} = \{\mathbf{A} \in \Re^{M \times N}, \tau\mathbf{1} \in \Re^M\}$ for all data samples, with a constant tresholding parameter $\tau$. The transform matrix $\mathbf{A}$ is one and same for the $C$ training and the testing models.

In the following section we describe the non-linear transform models, the proposed discriminative prior and its mesure, together with the main reason behind this particular use of non-linear transform models for learning and a sparsifying transform model for testing.

### 2.2 THE NON-LINEAR TRANSFORM MODEL WITH A DISCRIMINATIVE PRIOR

**The learning model** This paper defines a compact description of the non-linear transform (1) by a *non-linear transform model* as follows:

$$\mathbf{A}\mathbf{x}_{c,k} = \mathbf{y}_{c,k} + \mathbf{z}_{c,k}, \mathbf{y}_{c,k} = \mathcal{T}^{\mathcal{P}_c}(\mathbf{x}_{c,k}), \tag{2}$$

---

[3]That is the number of non-linear transforms equals the number of classes.

where $\mathcal{T}^{\mathcal{P}_c}(.) : \Re^N \to \Re^M$ is a parametric non-linear function that gives $\mathbf{y}_{c,k}$, by using the set of parameters $\mathcal{P}_c$. The term $\mathbf{z}_{c,k} = \mathbf{A}\mathbf{x}_{c,k} - \mathbf{y}_{c,k}$ is the *non-linear transform error* vector that represents the deviation of $\mathbf{A}\mathbf{x}_{c,k}$ from the targeted transform representation $\mathbf{y}_{c,k} = \mathcal{T}^{\mathcal{P}_c}(\mathbf{x}_{c,k})$ in the transform domain. Since the transform representation $\mathbf{y}_{c,k} = \mathcal{T}^{\mathcal{P}_c}(\mathbf{x}_{c,k})$ takes into acount a non-linearity, $\mathbf{A}\mathbf{x}_{c,k}$ is only seen as its linear approximation. Knowing something in advance about the difference between $\mathbf{y}_{c,k} - \mathbf{A}\mathbf{x}_{c,k}$ can be used in our model. However, since in advance we do not have any prior we assume that it is Gausssian like distributed. Therefore, the prior on $\mathbf{z}_{c,k}$ is modeled as $p(\mathbf{x}_{c,k}|\mathbf{y}_{c,k}, \mathbf{A}) \propto \exp(-\frac{\|\mathbf{A}\mathbf{x}_{c,k}-\mathbf{y}_{c,k}\|_2^2}{\beta_0})$, where $\beta_0$ is a scaling parameter. Assuming additionally that the non-linear function $\mathcal{T}^{\mathcal{P}_c}(\mathbf{x}_{c,k})$ gives sparse $\mathbf{y}_{c,k}$, then we have the improper prior on $\mathbf{y}_{c,k}$, defined as $p(\mathbf{y}_{c,k}) \propto \exp(-\frac{\|\mathbf{y}_{c,k}\|_1}{\beta_1})$ where $\beta_1$ is a scaling parameter. This paper models the joint probability $p(\boldsymbol{\tau}_1, \boldsymbol{\tau}_2, ...., \boldsymbol{\tau}_C, \mathbf{y}_{c,k})$ as

$$p(\boldsymbol{\tau}_1, \boldsymbol{\tau}_2, ...., \boldsymbol{\tau}_C, \mathbf{y}_{c,k}) \propto \exp(-\frac{1}{\beta_2} \min_{1 \le c1 \le C} D(\mathbf{y}_{c,k}; \boldsymbol{\tau}_{c1}))p(\mathbf{y}_{c,k}), \tag{3}$$

and assumes that $(A_{iid}) : p(\boldsymbol{\tau}_1, \boldsymbol{\tau}_2, ...., \boldsymbol{\tau}_C) = \prod_c p(\boldsymbol{\tau}_c)$, where $\beta_2$ is a scaling parameter. By the assumption in subsection 2.1, $\boldsymbol{\tau}_{c1}$ are spread far apart in the transform domain. Therefore, a minimum over $D(\mathbf{y}_{c,k}; \boldsymbol{\tau}_{c1})$ ensures that $\mathbf{y}_{c,k}$ in the transform domain will be located to the closest $\boldsymbol{\tau}_{c1}$ w.r.t. to the mesure $D(\mathbf{y}_{c,k}; \boldsymbol{\tau}_{c1})$. Moreover, given the class label $c$ using (3) and $(A_{iid})$ the model for the discriminative prior reduces to $p(\boldsymbol{\tau}_c|\mathbf{y}_{c,k}) \propto \exp(-\frac{D(\mathbf{y}_{c,k};\boldsymbol{\tau}_c)}{\beta_2})$ where $D(\mathbf{y}_{c,k}; \boldsymbol{\tau}_c)$ is a parametric measure with parameter $\boldsymbol{\tau}_c$. Assuming that $D(\mathbf{y}_{c,k}; \boldsymbol{\tau}_c)$ is determined by a relation on the support intersection between $\mathbf{y}_{c,k}$ and $\boldsymbol{\tau}_c$ we propose the following definition of measure:

$$D(\mathbf{y}_{c,k}; \boldsymbol{\tau}_c) = \|\mathbf{y}_{c,k}^+ \odot \boldsymbol{\tau}_c^+\|_1 + \|\mathbf{y}_{c,k}^- \odot \boldsymbol{\tau}_c^-\|_1 + \|\mathbf{y}_{c,k} \odot \boldsymbol{\tau}_c\|_2^2, \tag{4}$$

where $\|\mathbf{y}_{c,k}^+ \odot \boldsymbol{\tau}_c^+\|_1 + \|\mathbf{y}_{c,k}^- \odot \boldsymbol{\tau}_c^-\|_1$ measures the similarity contribution on the support intersection using the positive and the negative components, $\mathbf{y}_{c,k}^+, \boldsymbol{\tau}_c^+$ and $\mathbf{y}_{c,k}^-, \boldsymbol{\tau}_c^-$, respectively, of $\mathbf{y}_{c,k}$ and $\boldsymbol{\tau}_c$ and $\|\mathbf{y}_{c,k} \odot \boldsymbol{\tau}_c\|_2^2$ measures the strength of the support intersection between $\mathbf{y}_{c,k}$ and $\boldsymbol{\tau}_c$. The true $p(\boldsymbol{\tau}_c)$ and $\boldsymbol{\tau}_c$ are not known and instead of estimating them explicitly, an approximation to $D(\mathbf{y}_{c,k}; \boldsymbol{\tau}_c)$ is considered based only on the concentrations of the similarity on the support intersection and the expected strength of the support intersection for the transform data.

**Non-parametric approximation** We propose an approximation by sum of two expectations. The first one is the expected similarity on the support intersection for the positive and negative component between all $\mathbf{y}_{c,k}$ and the coresponding sets of the transform representations $\mathbf{Y}_{\backslash c}$ that come from all classes $c1$ different from $c$, i.e., $c \ne c1$. The second is the expected strength on the support intersection between $\mathbf{y}_{c,k}$ and the set of transform representations $\mathbf{Y}_{\backslash c}$ that come from all classes $c1$ different from $c$, $c \ne c1$. We define the approximation as:

$$\sum_{c,k} D(\mathbf{y}_{c,k}; \boldsymbol{\tau}_c) \sim D_{\ell_1}^{\mathcal{P}}(\mathbf{X}) + S_{\ell_2}^{\mathcal{P}}(\mathbf{X}) \text{ where}$$

$$D_{\ell_1}^{\mathcal{P}}(\mathbf{X}) = \sum_{c=1}^{C} \sum_{k=1}^{K} \sum_{c1 \in \{\{1,2,...,C\} \backslash c\}} \sum_{k1=1}^{K} (\|\mathbf{y}_{c,k}^+ \odot \mathbf{y}_{c1,k1}^+\|_1 + \|\mathbf{y}_{c,k}^- \odot \mathbf{y}_{c1,k1}^-\|_1),$$

$$S_{\ell_2}^{\mathcal{P}}(\mathbf{X}) = \sum_{c=1}^{C} \sum_{k=1}^{K} \sum_{c1 \in \{\{1,2,...,C\} \backslash c\}} \sum_{k1=1}^{K} \|\mathbf{y}_{c,k} \odot \mathbf{y}_{c1,k1}\|_2^2, \tag{5}$$

we highlight that the transform represntations $\mathbf{y}_{c,k}$ used in the approximation $D_{\ell_1}^{\mathcal{P}}(\mathbf{X})$ and $S_{\ell_2}^{\mathcal{P}}(\mathbf{X})$ are the result of applying the sparsifying transform with parameter set $\mathcal{P}$ to the data samples $\mathbf{x}_{c,k}$. The $m$-th element $y_{c,k}^+(m)$ of $\mathbf{y}_{c,k}^+$ is defined as $y_{c,k}^+(m) = \max(y_{c,k}(m), 0)$ and similarly, $y_{c,k}^-(m) = \max(-y_{c,k}(m), 0), \forall m \in \mathcal{M}$. We also define the expected similarity using the positive and negative components of all $\mathbf{y}_{c,k}$ across the transform representations $\mathbf{Y}_{\backslash c}$ that come from the same classes $c$ as $D_{\ell_1,c}^{\mathcal{P}}(\mathbf{X}) = \sum_c \sum_{k=1}^{K} \sum_{k1 \in \{\{1,2,...,K\} \backslash k\}} \left( \|\mathbf{y}_{c,k}^+ \odot \mathbf{y}_{c,k1}^+\|_1 + \|\mathbf{y}_{c,k}^- \odot \mathbf{y}_{c,k1}^-\|_1 \right)$. If the measure $D_{\ell_1}^{\mathcal{P}}(\mathbf{X})$ is not used then the approximation (5) is most similar to the one proposed in Liu et al. (2016).

By (5) a link is established between the non-linear transform models and the sparsifying transform model, or more generaly, assuming $\mathbf{A}$ is known, between a parametric and a non-parametric modeling view. The terms $\frac{2}{((C-1)K)(CK)} \sum_{c1 \in \{\{1,2,...,C\} \setminus c\}} \sum_{k1=1}^{K} \mathbf{y}_{c1,k1}^{-}$, $\frac{2}{((C1)K)(CK)} \sum_{c1 \in \{\{1,2,...,C\} \setminus c\}} \sum_{k1=1}^{K} \mathbf{y}_{c1,k1}^{+}$ and $\frac{2}{((C1)K)(CK)} \sum_{c1,c1 \neq c} \sum_{k1} \mathbf{y}_{c1,k1} \odot \mathbf{y}_{c1,k1}$ are seen as finite sample estimates [4] [5] of the positive, negative components and the Hadamard square, $\boldsymbol{\tau}_c^-$, $\boldsymbol{\tau}_c^+$ and $\boldsymbol{\tau}_c \odot \boldsymbol{\tau}_c$, respectivly, for the unknown variable $\boldsymbol{\tau}_c$, $\forall c \in \mathcal{C}$.

Note that the Fisher discriminate constraint Yang et al. (2011b), the pairwise constraint Guo et al. (2013) and the support intersection constraint Liu et al. (2016) are all approximations of a discriminative prior. However, they all have specific assumptions on the distribution of the data representation in the transform domain. The advantage of using (5) is that the approximation is without any prior to the probability distributions $p(\boldsymbol{\tau}_c)$ and without any explicit assumption about the metric/measure, or space/manifold in the transform domain.

Given a training set $\mathbf{X}$, the learning perspective w.r.t. a discriminative property is to estimate a non-linear transform models $\mathcal{P}_c = \{\mathbf{A} \in \Re^{M \times N}, \boldsymbol{\tau}_c \in \Re^M\}$ that minimize the empirical expectation $\frac{1}{CK} \sum_{c,k} \min_{1 \leq c \leq C} D(\mathbf{y}_{c,k}; \boldsymbol{\tau}_c) \sim \mathbb{E}[\min_{1 \leq c \leq C} D(\mathbf{y}_{c,k}; \boldsymbol{\tau}_c)]$. Moreover, if the corresponding class labels for the training set $\mathbf{X}$ are given[6] then $\sum_{c,k} \min_{1 \leq c \leq C} D(\mathbf{y}_{c,k}; \boldsymbol{\tau}_c)$ equals to $\sum_{c,k} D(\mathbf{y}_{c,k}; \boldsymbol{\tau}_c)$ and exactly matches the empirical expectation $D_{\ell_1}^{\mathcal{P}}(\mathbf{X}) + S_{\ell_2}^{\mathcal{P}}(\mathbf{X})$.

**The testing model** Assume that the transform matrix $\mathbf{A}$ and the parameters $\boldsymbol{\tau}_1, \boldsymbol{\tau}_2, ..., \boldsymbol{\tau}_C$ are known, then given any data sample $\mathbf{x}_{c,k}$ the transform representaton $\mathbf{y}_{c,k}$ is estimated as:

$$\min_{\mathbf{y}_{c,k}} \|\mathbf{A}\mathbf{x}_{c,k} - \mathbf{y}_{c,k}\|_2^2 + \lambda_0 (\min_{1 \leq c1 \leq C} D(\mathbf{y}_{c,k}; \boldsymbol{\tau}_{c1})) + \lambda_1 \|\mathbf{y}_{c,k}\|_1, \tag{6}$$

which is euqialent to $\min_{1 \leq c1 \leq C} \left( \min_{\mathbf{y}_{c,k}} \|\mathbf{A}\mathbf{x}_{c,k} - \mathbf{y}_{c,k}\|_2^2 + \lambda_0 D(\mathbf{y}_{c,k}; \boldsymbol{\tau}_{c1}) + \lambda_1 \|\mathbf{y}_{c,k}\|_1 \right)$. Furtheremore, if we assume that the measures $D(\mathbf{y}_{c,k}; \boldsymbol{\tau}_{c1})$ are zero, *i.e.*, $D(\mathbf{y}_{c,k}; \boldsymbol{\tau}_{c1}) = 0$, then the discriminative prior is non-informative, in a sence that it has no influence in the models. Only then do the non-linear transforms reduce to the sparsifying transform model, since (6) reduce to:

$$\min_{\mathbf{y}_{c,k}} \|\mathbf{A}\mathbf{x}_{c,k} - \mathbf{y}_{c,k}\|_2^2 + \lambda_1 \|\mathbf{y}_{c,k}\|_1. \tag{7}$$

Considering the testing stage, we note that the result (5) sheds light on another view. Namely, the sparsifying transform model $\mathcal{P} = \{\mathbf{A} \in \Re^{M \times N}, \lambda\mathbf{1} \in \Re^M\}$ is also seen as an approximation to the models represented by a set of parameters $\mathcal{P}_c = \{\mathbf{A} \in \Re^{M \times N}, \boldsymbol{\tau}_c \in \Re^M\}$ with expected loss in the discriminative properties of the transform representations expressed by the similarity concentration measure $D_{\ell_1}^{\mathcal{P}}(\mathbf{X}) + S_{\ell_2}^{\mathcal{P}}(\mathbf{X})$. At the same time this measure can also be considered as an empirical risk Vapnik (1995) w.r.t. the discriminative properties, related to the generalization capabilities Vapnik (1995) and Mark (2010) of the sparsifying transform model. Note that the same model is also the simplest that approximates the non-linear transform models used in the learning stage.

## 2.3 THE LEARNING ALGORITHM

In summary, the used priors are:

$$p(\mathbf{x}_{c,k}|\mathbf{y}_{c,k}, \mathbf{A}) \propto \exp\left(-\frac{\|\mathbf{A}\mathbf{x}_{c,k} - \mathbf{y}_{c,k}\|_2^2}{\beta_0}\right)$$

$$p(\boldsymbol{\tau}_c, \mathbf{y}_{c,k}) = p(\boldsymbol{\tau}_c|\mathbf{y}_{c,k})p(\mathbf{y}_{c,k}) \propto \exp\left(-\frac{D(\mathbf{y}_{c,k}; \boldsymbol{\tau}_c)}{\beta_2}\right) \exp\left(-\frac{\|\mathbf{y}_{c,k}\|_1}{\beta_1}\right). \tag{8}$$

---

[4]Since $\sum_{c1 \in \{\{1,2,...,C\} \setminus c\}} \sum_{k1} (\|\mathbf{y}_{c,k}^+ \odot \mathbf{y}_{c1,k1}^+\|_1 + \|\mathbf{y}_{c,k}^- \odot \mathbf{y}_{c1,k1}^-\|_1) = \|\left(\sum_{c1 \in \{\{1,2,...,C\} \setminus c\}} \sum_{k1} \mathbf{y}_{c1,k1}^+\right) \odot \mathbf{y}_{c,k}^+\|_1 + \|\left(\sum_{c1 \in \{\{1,2,...,C\} \setminus c\}} \sum_{k1} \mathbf{y}_{c1,k1}^-\right) \odot \mathbf{y}_{c,k}^-\|_1$, $\boldsymbol{\tau}_c^- \sim \frac{2}{((C-1)K)(CK)} \sum_{c1 \in \{\{1,2,...,C\} \setminus c\}} \sum_{k1} \mathbf{y}_{c1,k1}^-$ and $\boldsymbol{\tau}_c^+ \sim \frac{2}{((C-1)K)(CK)} \sum_{c1 \in \{\{1,2,...,C\} \setminus c\}} \sum_{k1} \mathbf{y}_{c1,k1}^+$.

[5]Since $\sum_{c1 \in \{\{1,2,...,C\} \setminus c\}} \sum_{k1} \|\mathbf{y}_{c,k} \odot \mathbf{y}_{c1,k1}\|_2^2 = \left(\sum_{c1 \in \{\{1,2,...,C\} \setminus c\}} \sum_{k1} \mathbf{y}_{c1,k1} \odot \mathbf{y}_{c1,k1}\right)^T (\mathbf{y}_{c,k} \odot \mathbf{y}_{c,k})$, $\boldsymbol{\tau}_c \odot \boldsymbol{\tau}_c \sim \frac{2}{((C-1)K)(CK)} \sum_{c1 \in \{\{1,2,...,C\} \setminus c\}} \sum_{k1} \mathbf{y}_{c1,k1} \odot \mathbf{y}_{c1,k1}$.

[6]Note that if the labels are not given then the unsupervised case can also be addressed by using a likelihood measure between a sample and the rest of the available samples, with the possibility to be defined in the original or in the transform domain.

Additionally, we have a prior on $\mathbf{A}$ that penalizes the information loss in order to avoid trivially unwanted matrices $\mathbf{A}$, *i.e.*, matrices that have repeated or zero rows. The prior is defined as:

$$p(\mathbf{A}) \propto \exp(-\Omega(\mathbf{A})) = \exp\left(-\left(\frac{1}{\beta_3}\|\mathbf{A}\|_F^2 + \frac{1}{\beta_4}\|\mathbf{A}\mathbf{A}^T - \mathbf{I}\|_F^2 - \frac{1}{\beta_5}\log|\det\mathbf{A}^T\mathbf{A}|\right)\right), \quad (9)$$

where the $\|\mathbf{A}\|_F$ penalty helps regularize the scale ambiguity, the $\log|\det(\mathbf{A}^T\mathbf{A})|$ and $\|\mathbf{A}\|_F^2$ are functions of the singular values of $\mathbf{A}$ and together help regularize the conditioning of $\mathbf{A}$. Assuming that the expected coherence $\mu^2(\mathbf{A})$ between the rows $\mathbf{a}_m$ of $\mathbf{A}$ (*i.e.*, $\mathbf{A}^T = [\mathbf{a}_1, \mathbf{a}_2, ..., \mathbf{a}_M]$) is defined as $\mu^2(\mathbf{A}) = \frac{2}{M(M-1)}\sum_{m_1 \neq m_2}|\mathbf{a}_{m_1}\mathbf{a}_{m_2}^T|^2, \forall m_1, m_2 \in \{1, 2, .., M\}$. Then $\|\mathbf{A}\mathbf{A}^T - \mathbf{I}\|_F^2$ measures the expected coherence $\mu^2(\mathbf{A})$ and the $\ell_2$ norm for the rows of $\mathbf{A}$.

Note that the joint probability can be expressed as:

$$p(\mathbf{x}_{c,k}, \mathbf{y}_{c,k}, \boldsymbol{\tau}_c, \mathbf{A}) = p(\mathbf{x}_{c,k}, \mathbf{y}_{c,k}, \boldsymbol{\tau}_c|\mathbf{A})p(\mathbf{A}), \quad (10)$$

where $p(\mathbf{x}_{c,k}, \mathbf{y}_{c,k}, \boldsymbol{\tau}_c|\mathbf{A}) = p(\mathbf{x}_{c,k}|\mathbf{y}_{c,k}, \mathbf{A})p(\boldsymbol{\tau}_c, |\mathbf{y}_{c,k})p(\mathbf{y}_{c,k})$, since $p(\mathbf{x}_{c,k}|\mathbf{y}_{c,k}, \boldsymbol{\tau}_c, \mathbf{A}) = p(\mathbf{x}_{c,k}|\mathbf{y}_{c,k}, \mathbf{A})$. Given the available training data set $\mathbf{X}$, maximizing $p(\mathbf{x}_{c,k}, \mathbf{y}_{c,k}, \boldsymbol{\tau}_c, \mathbf{A})$ over $\mathbf{Y}$ and $\mathbf{A}$ is same as minimizing the following problem:

$$\min_{\mathbf{Y},\mathbf{A}} \|\mathbf{A}\mathbf{X} - \mathbf{Y}\|_2^2 + \sum_{c,k}\lambda_0 D(\mathbf{y}_{c,k}; \boldsymbol{\tau}_c) + \lambda_1\|\mathbf{y}_{c,k}\|_1 + \Omega(\mathbf{A}), \quad (11)$$

where $\{\lambda_0, \lambda_1\}$ are inversely proportional to the scaling parameters $\{\beta_2, \beta_1\}$. Note that the solution to (11) is not equivalent to the maximum a priory (MAP) solution, which is difficult to compute, as it involves integrating over the vectors $\mathbf{y}_{c,k}$. Considering the optimization perspective, the problem is not convex in the variables $(\mathbf{Y}, \mathbf{A})$. The proposed solution here is obtained by iteratively, marginally maximizing the probability $p(\mathbf{x}_{c,k}, \mathbf{y}_{c,k}, \boldsymbol{\tau}_c, \mathbf{A})$ over $\mathbf{Y}$ and $\mathbf{A}$ which is equivalent to maximizing the conditional densities $p(\mathbf{y}_{c,k}|\mathbf{x}_{c,k}, \boldsymbol{\tau}_c, \mathbf{A})$ and $p(\mathbf{A}|\mathbf{x}_{c,k}, \mathbf{y}_{c,k}, \boldsymbol{\tau}_c)$, respectively. Meaning that at one iterating step one of the variables $\mathbf{Y}$ or $\mathbf{A}$ is fixed and w.r.t. the other the problem (11) is minimized. The following describes the iterating steps that consist of linear map estimation (maximizing $p(\mathbf{A}|\mathbf{x}_{c,k}, \mathbf{y}_{c,k}, \boldsymbol{\tau}_c)$) and discriminative encoding (maximizing $p(\mathbf{y}_{c,k}|\mathbf{x}_{c,k}, \boldsymbol{\tau}_c, \mathbf{A})$).

**Linear map estimation**: Given the available data samples $\mathbf{X}$ and the corresponding transform representations $\mathbf{Y}$ the linear map $\mathbf{A}$ estimation problem reduces to:

$$\min_{\mathbf{A}} \|\mathbf{A}\mathbf{X} - \mathbf{Y}\|_2^2 + \frac{\lambda_2}{2}\|\mathbf{A}\|_F^2 + \frac{\lambda_3}{2}\|\mathbf{A}\mathbf{A}^T - \mathbf{I}\|_F^2 - \lambda_4\log|\det\mathbf{A}^T\mathbf{A}|, \quad (12)$$

where $\{\lambda_2, \lambda_3, \lambda_4\}$ are inversely proportional to the scaling parameters $\{\beta_3, \beta_4, \beta_5\}$ and we use the $\epsilon$-close closed form solution estimated as follows:

***Proposition* 1** ($\epsilon$-*close closed form solution*): *Given* $\mathbf{Y} \in \Re^{M \times CK}$, $\forall\mathbf{X} \in \Re^{N \times CK}$ *and* $M \geq N$, $\forall\lambda_2 \geq 0, \lambda_3 \geq 0$ *and* $\lambda_4 \geq 0$ *let the eigen value decomposition* $\mathbf{U}_X\boldsymbol{\Sigma}_X\mathbf{V}_X^T$ *of* $\mathbf{X}\mathbf{X}^T + \lambda_2\mathbf{I}$ *and the singular value decomposition* $\mathbf{U}_{U_XXY}\boldsymbol{\Sigma}_{U_XXY}\mathbf{V}_{U_XXY}^T$ *of* $\mathbf{U}_X^T\mathbf{X}\mathbf{Y}^T$ *exist, then if and only if* $\sigma_X(n) > 0, \forall n \in N = \{1, 2, 3, ..., N\}$, *(12) has* $\epsilon$-*close approximative solution as*:

$$\mathbf{A} = \mathbf{V}_{U_XXY}\mathbf{U}_{U_XXY}^T\boldsymbol{\Sigma}_A\boldsymbol{\Sigma}_X^{-1}\mathbf{U}_X^T, \quad (13)$$

*where* $\boldsymbol{\Sigma}_A$ *is diagonal matrix*, $\Sigma_A(n, n) = \sigma_A(n) \geq 0$, *and* $\sigma_A(n)$ *are solutions to quartic polynomials with global minimums (the proof is given in Appendix A.2).*

**Discriminative encoding**: Given the available data samples $\mathbf{X}$ and the current estimate of the transform $\mathbf{A}$ the discriminative representation estimation problem is formulated as $(P_{DR})$ : $\min_{\mathbf{Y}}\|\mathbf{A}\mathbf{X} - \mathbf{Y}\|_F^2 + \lambda_0\sum_{c,k}D(\mathbf{y}_{c,k}; \boldsymbol{\tau}_c) + \lambda_1\|\mathbf{y}_{c,k}\|$. $(P_{DR})$. Even not knowing $p(\boldsymbol{\tau}_c)$ or the model variables $\boldsymbol{\tau}_c$ we show that by the approximation (5), $(P_{DR})$ has an efficient solution. Assuming that $\mathbf{Y}_{\backslash c}$ is given, using the approximation (5), then for any sample $k \in \mathcal{K}$ from any class $c \in \mathcal{C}$, problem $(P_{DR})$ reduces to a *constrained projection* problem:

$$\min_{\mathbf{y}_{c,k}} \|\mathbf{A}\mathbf{x}_{c,k} - \mathbf{y}_{c,k}\|_2^2 + \lambda_0\left(\mathbf{g}_c^T|\mathbf{y}_{c,k}| + \mathbf{s}_c^T(\mathbf{y}_{c,k} \odot \mathbf{y}_{c,k})\right) + \lambda_1\mathbf{1}^T|\mathbf{y}_{c,k}|, \quad (14)$$

and has a closed form solution as:

$$\mathbf{y}_{c,k} = \text{sign}(\mathbf{A}\mathbf{x}_{c,k}) \odot \max(|\mathbf{A}\mathbf{x}_{c,k}| - \lambda_0\mathbf{g}_c - \lambda_1\mathbf{1}, \mathbf{0}) \oslash (\mathbf{1} + 2\lambda_0\mathbf{s}_c), \quad (15)$$

where we abuse notation to denote $|\mathbf{y}_{c,k}|$ as the vector having as elements the absolute values of the corresponding elements in $\mathbf{y}_{c,k}$, $\oslash$ is a Haddamard division, $\mathbf{g}_c = \text{sign}(\max(\mathbf{A}\mathbf{x}_{c,k}, \mathbf{0})) \odot \mathbf{d}_c^+ + \text{sign}(\max(-\mathbf{A}\mathbf{x}_{c,k}, \mathbf{0})) \odot \mathbf{d}_c^-$, $\mathbf{d}_c^+ = \sum_{\substack{c1 \\ c1 \neq c}} \sum_{k1} \mathbf{y}_{c1,k1}^+$, $\mathbf{d}_c^- = \sum_{\substack{c1 \\ c1 \neq c}} \sum_{k1} \mathbf{y}_{c1,k1}^-$ and $\mathbf{s}_c = \sum_{\substack{c1 \\ c1 \neq c}} \sum_{k1} \mathbf{y}_{c1,k1} \odot \mathbf{y}_{c1,k1}$ (*the proof is given in Appendix B*).

We note that at convergence (which we do not prove here) we can only claim that a joint local maximum in $(\mathbf{Y}, \mathbf{A})$ of $p(\mathbf{x}_{c,k}, \mathbf{y}_{c,k}, \boldsymbol{\tau}_c, \mathbf{A})$ has been reached, even if, as in this case, each optimization step achieves the (marginal) $\epsilon$-close and the global optimal solution, respectively.

## 2.4 A MEASURE FOR THE DISCRIMINATIVE PROPERTIES AND ITS BOUNDS

This paper proposes a notion for the discriminative properties of a data set under a non-linear transform named as discrimination power, based on a measure for the relations between the concentrations $D_{\ell_1,c}^{\mathcal{P}}(\mathbf{X})$ and $D_{\ell_1}^{\mathcal{P}}(\mathbf{X})$.

**Proposition 2**: *The discrimination power for any dataset* $\mathbf{Y} \in \Re^{M \times CK}$ *under a non-linear transform with parameter set* $\mathcal{P}$ *is defined as:*

$$\mathcal{I}^t = \log(D_{\ell_1,c}^{\mathcal{B}^M}(\mathbf{Y})) - \log(D_{\ell_1}^{\mathcal{B}^M}(\mathbf{Y}) + \epsilon) = \log(D_{\ell_1,c}^{\mathcal{P}}(\mathbf{X})) - \log(D_{\ell_1}^{\mathcal{P}}(\mathbf{X}) + \epsilon), \quad (16)$$

*where* $\mathcal{B}^M = \{\mathbf{A}_t \in \Re^{M \times M}, \mathbf{0} \in \Re^M\}$, $\mathbf{A}_t = \mathbf{I}$ *and* $\epsilon > 0$ *is a small positive constant.*

**Remark 1**: *The advantage of this measure is that it logarithmically signifies the difference between* $D_{\ell_1,c}^{\mathcal{P}}(\mathbf{X})$ *and* $D_{\ell_1}^{\mathcal{P}}(\mathbf{X})$[7] .

The definition about the discrimination power of the original data set $\mathbf{X}$, but, now under a model with a parameter set $\mathcal{B}^N = \{\mathbf{A}_o \in \Re^{N \times N}, \mathbf{0} \in \Re^N\}$ where $\mathbf{A}_o = \mathbf{I}$ is equivalent to the one defined for $\mathcal{I}^t$. We denote it as $\mathcal{I}^o$. The bound on the discrimination power is given by the following result.

**Theorem 1**: *The discrimination power for any data set* $\mathbf{X} \in \Re^{N \times CK}$ *under any non-linear transform with parameter set* $\mathcal{P}$ *is bounded as*:

$$\log(\lambda_{min}(\mathbf{A}^T\mathbf{A})) + \log\left(\frac{Tr\{\frac{\partial D_{\ell_1,c}^{\mathcal{B}^N}(\mathbf{X})}{\partial \mathbf{A}_o}|_{\mathbf{A}_o=\mathbf{I}}\}}{D_{\ell_1}^{\mathcal{B}^M}(\mathbf{A}\mathbf{X}) + \epsilon}\right) \leq \mathcal{I}^t \leq \log\left(D_{\ell_1,c}^{\mathcal{B}^M}(\mathbf{A}\mathbf{X})\right) - \log \epsilon. \quad (17)$$

*The proof is given in Appendix D*[8].

At first the resulting bounds might look counterintuitive since the loss of information seems to increase the discrimination power. This fact is true, however, up to a certain limit. Therefore, it is important to distinguish two main conclusions. First, for any model with a set of parameters $\mathcal{P}$ for which there is no loss of information, that is, no thresholding, the only condition for the increase in the discrimination power is $D_{\ell_1}^{\mathcal{B}^N}(\mathbf{X}) \geq D_{\ell_1}^{\mathcal{B}^M}(\mathbf{A}\mathbf{X})$ and $D_{\ell_1,c}^{\mathcal{B}^N}(\mathbf{X}) \leq D_{\ell_1,c}^{\mathcal{B}^M}(\mathbf{A}\mathbf{X})$. Second, in the rest of the cases for which $D_{\ell_1}^{\mathcal{B}^N}(\mathbf{X}) \geq D_{\ell_1}^{\mathcal{B}^M}(\mathbf{Y})$ and $D_{\ell_1,c}^{\mathcal{B}^N}(\mathbf{X}) \leq D_{\ell_1,c}^{\mathcal{B}^M}(\mathbf{Y})$ holds true it will be possible to increase the discrimination power. Moreover, there is a trade-off between the increase in discrimination power as a result of the loss of information as consequence of the non-linear thresholding operation.

---

[7]Assume we have a discriminative prior defined as $p(\boldsymbol{\tau}_c|\mathbf{y}_{c,k}) \propto \exp(-\frac{\|\boldsymbol{\tau}_c^+ \odot \mathbf{y}_{c,k}^+\|_1 + \|\boldsymbol{\tau}_c^- \odot \mathbf{y}_{c,k}^-\|_1}{\beta_2})$ and $p(\boldsymbol{\upsilon}_c|\mathbf{y}_{c,k}) \propto \exp(-\frac{\|\boldsymbol{\upsilon}_c^+ \odot \mathbf{y}_{c,k}^+\|_1 + \|\boldsymbol{\upsilon}_c^- \odot \mathbf{y}_{c,k}^-\|_1}{\beta_2})$, where $\boldsymbol{\tau}_c$ and $\boldsymbol{\upsilon}_c$ are unknown parameters. Then the difference $D_{\ell_1,c}^{\mathcal{P}}(\mathbf{X}) - D_{\ell_1}^{\mathcal{P}}(\mathbf{X})$ between $D_{\ell_1,c}^{\mathcal{P}}(\mathbf{X})$ and $D_{\ell_1}^{\mathcal{P}}(\mathbf{X})$ actually represents a finite sample approximation to a *discriminative density* since it approximates the density $\log\left(\frac{p(\boldsymbol{\tau}_c|\mathbf{y}_{c,k})}{p(\boldsymbol{\upsilon}_c|\mathbf{y}_{c,k})}\right)$, *i.e.*, $D_{\ell_1,c}^{\mathcal{P}}(\mathbf{X}) - D_{\ell_1}^{\mathcal{P}}(\mathbf{X}) \sim -\log\left(\frac{p(\boldsymbol{\tau}_c,\mathbf{y}_{c,k})}{p(\boldsymbol{\upsilon}_c,\mathbf{y}_{c,k})}\right)$ and $-\log\left(\frac{p(\boldsymbol{\tau}_c,\mathbf{y}_{c,k})}{p(\boldsymbol{\upsilon}_c,\mathbf{y}_{c,k})}\right) = D(\boldsymbol{\tau}_c,\mathbf{y}_{c,k}) - D(\boldsymbol{\upsilon}_c,\mathbf{y}_{c,k}) = \|\boldsymbol{\tau}_c^+ \odot \mathbf{y}_{c,k}^+\|_1 + \|\boldsymbol{\tau}_c^- \odot \mathbf{y}_{c,k}^-\|_1 - (\|\boldsymbol{\upsilon}_c^+ \odot \mathbf{y}_{c,k}^+\|_1 + \|\boldsymbol{\upsilon}_c^- \odot \mathbf{y}_{c,k}^-\|_1)$.

[8]In *Appendix C* we also provide a sensitivity analysis that complements our result since it is related to the notion about the discriminative quality of the representation. The result in *Appendix C* also gives an information-theoretic interpretation and information-geometric perspective about the model and the similarity concentration measure without the need of strict conditions for regularity, i.e., smoothness of the manifolds.

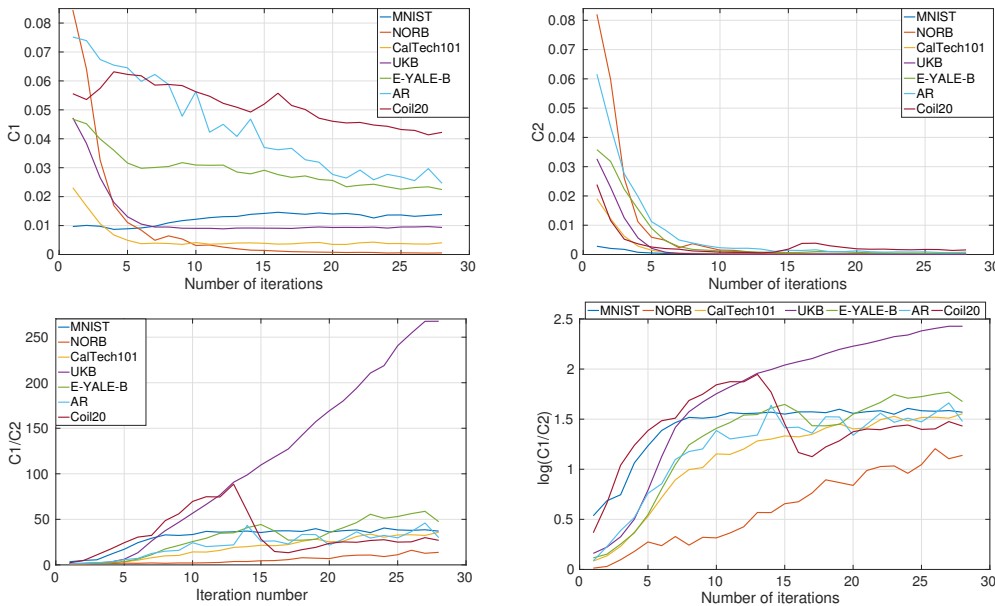

Figure 2: The evolution of the similarity concentrations $C1 = D_{\ell_1,c}^{\mathcal{B}^M}(\mathbf{Y})$ and $C2 = D_{\ell_1}^{\mathcal{B}^M}(\mathbf{Y})$, their ratio $C1/C2$ and the discrimination power $\log(C1/C2) = \mathcal{I}^t$ during the learning of the non-linear transform with transform dimension $M = 19000$.

|       | $C_n(\mathbf{A})$ | $\mu(\mathbf{A})$ | $t_e[min]$ | | $\mathcal{I}^O$ | $\mathcal{I}^{RT}$ | $\mathcal{I}^{ST^*}$ | $\mathcal{I}^{NT}$ |
|-------|-------------------|-------------------|------------|---|-----------------|--------------------|----------------------|--------------------|
| $D1$  | 2.21              | 0.03              | 5.10       | | 0.03            | 0.18               | 0.68                 | **1.98**           |
| $D2$  | 1.80              | 0.02              | 5.45       | | 0.02            | 0.10               | 1.30                 | **1.79**           |
| $D3$  | 2.12              | 0.02              | 6.55       | | 0.00            | 0.01               | 0.71                 | **1.61**           |
| $D4$  | 0.08              | 0.02              | 8.92       | | 0.08            | 0.61               | 0.89                 | **1.89**           |
| $D5$  | 6.01              | 0.01              | 12.8       | | 0.01            | 0.16               | 1.02                 | **2.12**           |
| $D6$  | 33.1              | 0.02              | 30.1       | | 0.06            | 0.53               | 1.36                 | **3.36**           |
| $D7$  | 1.60              | 0.02              | 5.00       | | 0.13            | 0.63               | 1.06                 | **1.96**           |

Table 1: The conditioning number $C_n(\mathbf{A}) = \frac{\lambda_{max}}{\lambda_{min}}$ and the expected mutual coherence $\mu(\mathbf{A})$ for the learned transform $\mathbf{A}$. The execution time $t_e[min]$ in minutes of the proposed algorithm for 28 iterations at the transform domain dimensionality $M = 19000$.

## 3 NUMERICAL EXPERIMENTS

The numerical experiments are summarized in two different parts. In the first series of the experiments the properties of the learned map $\mathbf{A}$ for the proposed algorithm are investigated. We evaluate the computational efficiency, as run time $t_e[min]$, the conditioning number $C_n(\mathbf{A}) = \frac{\lambda_{max}}{\lambda_{min}}$, the expected mutual coherence $\mu(\mathbf{A})$ and the discrimination power across several databases for a learned non-linear transforms having different dimensionality. A comparison between the discrimination power uder different transforms is presented. The discriminatation power is estimated in the original domain, after transform by a random matrix (having Gaussian random samples as entries and transform dimension of $M = 19000$) and after a learned non-linear transform having transform dimension $M = 19000$ without and with discriminative prior, denoted as $\mathcal{I}^0, \mathcal{I}^{RT}, \mathcal{I}^{ST^*}$ and $\mathcal{I}^{NT}$, respectively. The second part evaluates a comparison of the discrimination power between the proposed algorithm and different DDL methods Ramirez et al. (2010), Yang et al. (2011a), Vu et al. (2015) and Vu & Monga (2016). This comparison considers a setup where the used data sets are divided into a training and test set. Moreover, the learning is performed on the training set and the evaluation is performed on the test set. In the same series of experiments the recognition accuracy for the two data sets is also computed and compared.

**Data sets and algorithms set up** The used data sets are Extended YALE B ($D1$)Georghiades et al. (2001), AR ($D2$) Martínez & Benavente (1998), Norb ($D3$) LeCun et al. (2004), Coil-20 ($D4$)

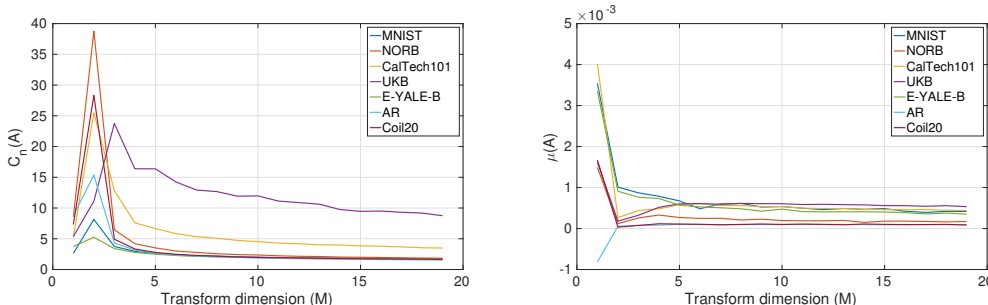

Figure 3: The conditioning number $C_n(\mathbf{A}) = \frac{\lambda_{max}}{\lambda_{min}}$ and the expected mutual coherence $\mu(\mathbf{A})$ for the learned linear transform $\mathbf{A}$ at different dimensionality $M \in \mathcal{Q}$.

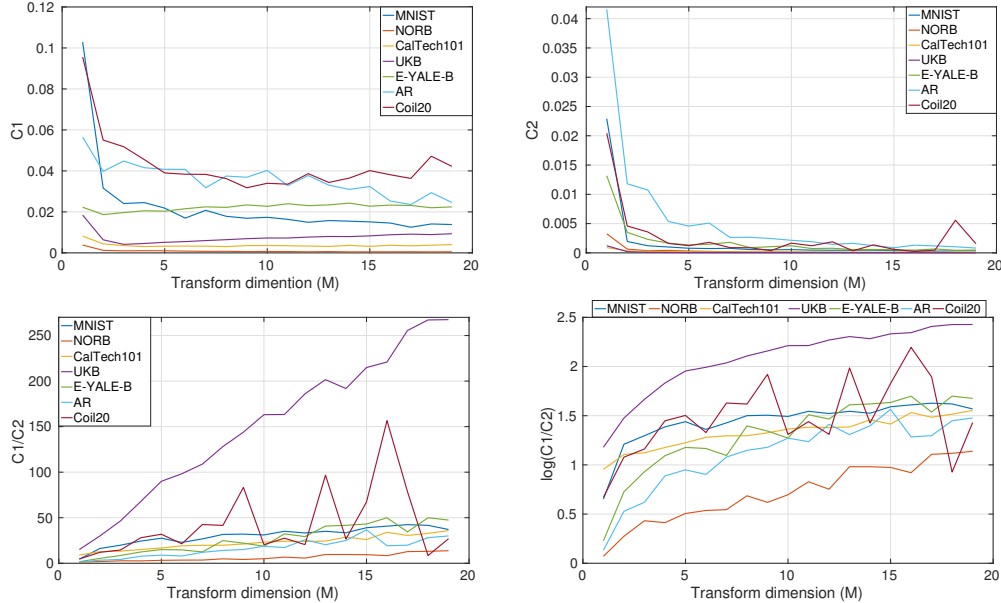

Figure 4: The similarity concentrations $C1 = D_{\ell_1,c}^{\mathcal{B}^M}(\mathbf{Y})$ and $C2 = D_{\ell_1}^{\mathcal{B}^M}(\mathbf{Y})$, their ratio $C1/C2$ and the discrimination power $\log(C1/C2) = \mathcal{I}^t$ on a subset of the transform data using learned non-linear transform at different dimensionality $M \in \mathcal{Q}$.

Nene et al. (1996), Clatech101 ($D5$) LeCun et al. (2008), UKB ($D6$) Nistér & Stewénius (2006) and MNIST ($D7$) Lecun & Cortes. All the images from the respective datasets were downscaled to resolutions $21 \times 21$, $32 \times 28$, $24 \times 24$, $20 \times 25$, $21 \times 21$, $20 \times 25$, $28 \times 28$, respectively, and are normalized to unit variance. Considering the used implementation of the algorithm we note that the singular value decomposition for a large matrix has high computational complexity. However, $\mathbf{A} - \hat{\mathbf{A}}$, where $\hat{\mathbf{A}}$ is estimated as a solution in the transform update step, can be considered as an proximal operator Parikh & Boyd (2014) for the gradient of the objective (12). Additionally, instead of using all of the available data samples $\mathbf{X}$, a subset of them might be used. Therefore, one simple on-line variant for the update of $\mathbf{A}$ w.r.t. a subset of the available training set has the form $\mathbf{A}^{t+1} = \mathbf{A}^t - \rho(\mathbf{A}^t - \hat{\mathbf{A}}^t)$ with $\rho$ a predefined step size. In the numerical experiments we use the on-line variant of the algorithm (the convergence analysis for this variant of the algorithm is left for future work) were we used a baches of sizes equal to $10\% - 12\%$ of the total amount of the available training data. The parameters $\lambda_0$ and $\lambda_1$ are set such that the resulting non-linear transform representation has a very small number of non-zeros w.r.t. the transform dimension. In the experiments this number is set to be 15. The rest of the parameters are set as $\{\lambda_2, \lambda_3, \lambda_4\} = \{1000000, 1000000, 1000000\}$. The algorithm is initialized with a random matrix having i.i.d. Gaussian (zero mean, variance one) entries and is terminated after the 28th iteration. The results are obtained as the average of 3 runs. An implementation presented in Vu & Monga (2016) was used to learn the dictionaries and estimate the

| | $D1$ | $D7$ | | $D1$ Acc. [%] | | $D7$ Acc. [%] |
|---|---|---|---|---|---|---|
| $\mathcal{I}^{DLSI}$ | 0.71 | 0.67 | $DLSI$ | 96.5 | $DLSI$ | 98.74 |
| $\mathcal{I}^{FDDL}$ | 0.87 | 0.63 | $FDDL$ | 97.5 | $FDDL$ | 96.31 |
| $\mathcal{I}^{COPAR}$ | 0.57 | 0.54 | $COPAR$ | 98.3 | $COPAR$ | 96.41 |
| $\mathcal{I}^{LRSDL}$ | 0.42 | 0.40 | $LRSDL$ | 98.7 | $LRSDL$ | – |
| $\mathcal{I}^{NT}$ | **0.98** | **0.81** | $NT$ | **99.7** | $NT$ | **99.02** |
| | a) | | | b) | | c) |

Table 2: a) The discrimination power for the methods $DLSI$ Ramirez et al. (2010), $FDDL$ Yang et al. (2011a), $COPAR$ Vu et al. (2015) and $LRSDL$ Vu & Monga (2016) and the proposed method $NT$, b) and c) The recognition results on the Extended Yale B and MNIST database.

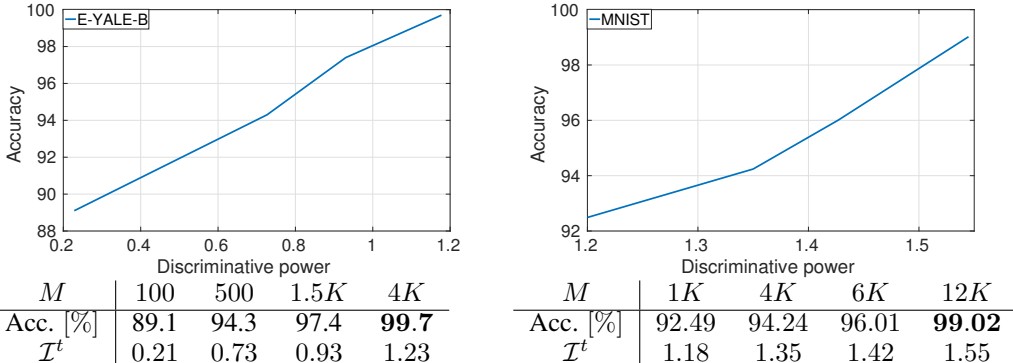

| $M$ | 100 | 500 | $1.5K$ | $4K$ |
|---|---|---|---|---|
| Acc. [%] | 89.1 | 94.3 | 97.4 | **99.7** |
| $\mathcal{I}^t$ | 0.21 | 0.73 | 0.93 | 1.23 |

| $M$ | $1K$ | $4K$ | $6K$ | $12K$ |
|---|---|---|---|---|
| Acc. [%] | 92.49 | 94.24 | 96.01 | **99.02** |
| $\mathcal{I}^t$ | 1.18 | 1.35 | 1.42 | 1.55 |

Figure 5: The recognition results and the discrimination power on the Extended Yale B and MNIST databases, respectively, using a non-linear transform with different dimensionality $M$ and linear SVM classifier on top of the transform representation.

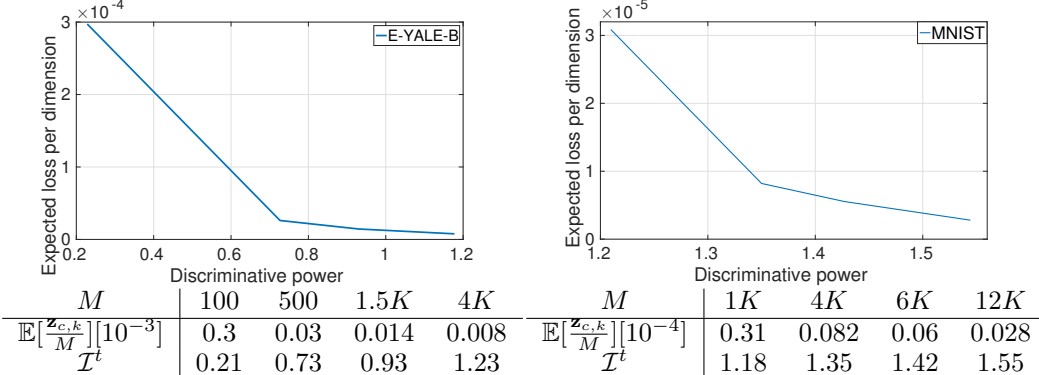

| $M$ | 100 | 500 | $1.5K$ | $4K$ |
|---|---|---|---|---|
| $\mathbb{E}[\frac{\mathbf{z}_{c,k}}{M}][10^{-3}]$ | 0.3 | 0.03 | 0.014 | 0.008 |
| $\mathcal{I}^t$ | 0.21 | 0.73 | 0.93 | 1.23 |

| $M$ | $1K$ | $4K$ | $6K$ | $12K$ |
|---|---|---|---|---|
| $\mathbb{E}[\frac{\mathbf{z}_{c,k}}{M}][10^{-4}]$ | 0.31 | 0.082 | 0.06 | 0.028 |
| $\mathcal{I}^t$ | 1.18 | 1.35 | 1.42 | 1.55 |

Figure 6: The expected loss $\mathbb{E}[\|\frac{\mathbf{z}_{c,k}}{M}\|_2^2] = \mathbb{E}[\frac{\|\mathbf{A}\mathbf{x}_{c,k} - \mathbf{y}_{c,k}\|_2^2}{M}]$ and the discrimination power on the Extended Yale B and MNIST databases, respectively, on the transform representation $\mathbf{Y}$, obtained by using a non-linear transform $\mathcal{T}^{\mathcal{P}}$ at different dimensionality $M$.

sparse codes for the respective supervised dictionary learning methods DDL Ramirez et al. (2010), Yang et al. (2011a), Vu et al. (2015) and Vu & Monga (2016).

**Linear map properties, the similarity concentrations and the discrimination power** The conditioning number and the expected coherence for the learned transforms are shown on Table 1. The learned transforms for all the databases have good conditioning numbers and low expected coherence. The running time $t_e$, measured in minutes, and the number of used dimensions, denoted as $M$ are also shown in Table 1. The learned transforms for all the data sets have relatively low execution time, regardless of the very high transform dimension $M = 19000$. The discrimination power is significantly increased in the transform domain $\mathcal{I}^{NT}$ compared to the one in the original domain $\mathcal{I}^O$ and is higher than $\mathcal{I}^{ST^*}$ and $\mathcal{I}^{RT}$. The evolution of the similarity concentrations $C1 = D_{\ell_{1,c}}^{\mathcal{B}^M}(\mathbf{Y})$ and $C2 = D_{\ell_1}^{\mathcal{B}^M}(\mathbf{Y})$, their ratio $C1/C2$ and the discrimination power $\log(C1/C2) = \mathcal{I}^t$ for subsets of the used databases after applying a non-linear transform with transform dimension $M = 19000$ is shown in Figure 2. It is important to note that the similarity concentrations $C1 = D_{\ell_{1,c}}^{\mathcal{B}^M}(\mathbf{Y})$ and

$C2 = D_{\ell_1}^{\mathcal{B}^M}(\mathbf{Y})$ are decreasing, meaning that there is a loss of information. However how this loss effects the resulting similarity concentration is crucial for the discrimination properties. As shown in Figure 2, the slope of decrease for $C2$ is stronger. Therefore, the discrimination power increases per iteration. For the Coil-20 ($D4$) database there is a fluctuation. This is explained by the fact that during learning we used a small number of data samples from the same database and that in the data there is high variability. The conditioning number and the expected coherence for the learned transforms for all the databases at different transform dimensions $M \in \mathcal{Q} = \{100, 1150, 2200, 3250, 4300, 5350, 6400, 7450, 8500, 9550, 10600, 11650, 12700, 13750, 14800, 15850, 16900, 17950, 19000\}$ are shown in Figure 3. We see that the value of both the conditioning number and the coherence is reducing and it converges to common values. This confirms the effectiveness of the conditioning and the coherence constraints. The similarity concentrations $C1 = D_{\ell_1,c}^{\mathcal{B}^M}(\mathbf{Y})$ and $C2 = D_{\ell_1}^{\mathcal{B}^M}(\mathbf{Y})$, their ratio $C1/C2$ and the discrimination power $\log(C1/C2) = \mathcal{I}^t$ for a subsets of the used databases after applying a non-linear transform having transform dimensions $M \in \mathcal{Q}$ is shown in Figure 4. We can see similar behavior as previous, that is, $C1$ and $C1$ are decreasing, but, the slope of decrease for $C2$ is stronger. Therefore, the discrimination power increases as the transform dimension increases.

**NT vs DDL discrimination power and recognition performance** The proposed method is compared with $DLSI$Ramirez et al. (2010), $FDDL$ Yang et al. (2011a), $COPAR$ Vu et al. (2015) and $LRSDL$ Vu & Monga (2016). Half of the data samples from the data set Extended YALE B, sampled at random are used for learning and the remaining other half are used for evaluation. Considering the MNIST database the training set is used for learning and the test set is used for both evaluating the discrimination power and the recognition accuracy. We compute both the discrimination power and the recognition accuracy on a subsets from the test sets. The dictionary size (transform dimension $M$) is set to be equal to $\{150, 75, 1515, 3825, 570, 150, 300\}$for the used databases, respectivly, in all of the comparing algorithms. The discrimination power of the comparing methods is denoted as $\mathcal{I}^{DLSI}, \mathcal{I}^{FDDL}, \mathcal{I}^{COPAR}$ and $\mathcal{I}^{LRSDL}$. The recognition results for the methods $DLSI$, $FDDL, COPAR$ and $LRSDL$ on the data sets Extended YALE B and MNIST were not computed here, rather we use the best reported result form the respective papers Ramirez et al. (2010), Yang et al. (2011a), Vu et al. (2015) and Vu & Monga (2016). Considering the proposed algorithm the non-linear transform was learned for the transform dimensions $M = \{100, 500, 1500, 4000\}$ and $M = \{1000, 4000, 6000, 12000\}$, respectively, for the used data sets. After the transform was learned, the transform data samples were computed for the respective training and test sets. Then, the transform training data samples were used as features to learn a linear SVM classifier in one-against-all regime. The results are shown in Table 2 a), b) and c). The discrimination power of the proposed non-linear transform is higher that the discrimination power of the comparing methods. The recognition accuracy is higher for high dimensionality of the proposed method and outperforms the DDL methods at dimensionality 4000 and 12000. In Figure 5 and Figure 6 are shown the recognition accuracy and the expected loss measured as $\mathbb{E}[\|\frac{\mathbf{z}_{c,k}}{M}\|_2^2] = \mathbb{E}[\frac{\|\mathbf{A}\mathbf{x}_{c,k} - \mathbf{y}_{c,k}\|_2^2}{M}]$ as a linear function of the discrimination power evaluated at transform dimension $M = \{100, 500, 1500, 4000\}$ and $M = \{1000, 4000, 6000, 12000\}$. It is interesting to highlight that as the discrimination power at different transform dimension increases it also increases the accuracy of recognition. Moreover, the results on these two data sets show that this increase is approximately linear. On the other hand the expected loss decreases as the discrimination power at different transform dimensions increases.

## 4 CONCLUSION

This paper presented a novel approach for learning discriminative and sparse representations. A novel discriminative prior was proposed and the properties of the models with the prior were investigated. A low complexity learning algorithm was presented. The preliminary results w.r.t. the introduced measures and the recognition accuracy on the used databases showed promising performance. We showed that it is possible to increase the discrimination power with information loss. Moreover, we highlight that when expanding to high dimensional space with non-linear transforms how the loss of information reflects the similarity concentrations is crucial for the discriminative properties. A study on the recognition capabilities for other databases are our next steps. An extention considering the sufficient conditions for increase in discrimination power in the transform domain, under supervised and unsupervised case, together with an analysis for a deep architecture where per single layer we have a non-linear transforms are left for our future work.

APPENDIX A.

A.1 THE GLOBAL OPTIMAL SOLUTION

Given $\mathbf{X}$ and the curent estimate of $\mathbf{Y}$, the estimate of the transform is a solution to the following problem

$$\min_{\mathbf{A}} \|\mathbf{AX} - \mathbf{Y}\|_2^2 + \frac{\lambda_2}{2}\|\mathbf{A}\|_F^2 + \frac{\lambda_3}{2}\|\mathbf{AA}^T - \mathbf{I}\|_F^2 - \lambda_4 \log|\det \mathbf{A}^T\mathbf{A}|. \tag{18}$$

**Theorem 2** (golobal optimal solution) *Given* $\mathbf{X} \in \Re^{N \times CK}$ *and* $\mathbf{Y} \in \Re^{M \times CK}$*, if and only if the joint decomposition*

$$\begin{aligned} \mathbf{XX}^T &= \mathbf{U}_X \mathbf{\Sigma}_X^2 \mathbf{U}_X^T \\ \mathbf{XY}^T &= \mathbf{U}_X \mathbf{\Sigma}_{XY} \mathbf{V}_{XY}^T, \end{aligned} \tag{19}$$

*exists, where* $\mathbf{U}_X \in \Re^{N \times N}$ *is orthonormal,* $\mathbf{V}_{XY} \in \Re^{M \times N}$ *is per columns orthonormal and* $\mathbf{\Sigma}_X, \mathbf{\Sigma}_{XY} \in \Re^{N \times N}$ *are diagonal matrices with positive diagonal elements, then* (18) *has a global minimum as*

$$\mathbf{A} = \mathbf{V}_{XY} \mathbf{\Sigma}_A \mathbf{\Sigma}_X^{-1} \mathbf{U}_X^T, \tag{20}$$

$\Sigma_A(n,n) = \sigma_A(n), \forall n, \sigma_A(n) \geq 0$ *and* $\sigma_A(n)$ *are positive solutions to*

$$\frac{\lambda_3}{\sigma_X^4(n)}\sigma_A^4(n) + \frac{\sigma_X^2(n) - 2\lambda_3}{\sigma_X^2(n)}\sigma_A^2(n) - \frac{\sigma_{XY}(n)}{\sigma_X(n)}\sigma_A(n) - 2\lambda_4 \log\frac{\sigma_X(n)}{\sigma_A(n)} = 0. \tag{21}$$

**Proof of Theorem 2** Consider the equvalent trace form of (18)

$$\begin{aligned} \min_{\mathbf{A}} Tr\{(\mathbf{AX} - \mathbf{Y})^T \mathbf{AX} - \mathbf{Y}\} + \lambda_2 Tr\{\mathbf{A}^T\mathbf{A}\} + \\ \lambda_3 Tr\{(\mathbf{AA}^T - \mathbf{I})^T(\mathbf{AA}^T - \mathbf{I})\} - \lambda_4 \log|\mathbf{A}^T\mathbf{A}|. \end{aligned} \tag{22}$$

Note that since $\lambda_2 \geq 0, \mathbf{XX}^T + \lambda_2\mathbf{I}$ is a symetric positive definite matrix whit all eigenvalues non-negative, therfore it decomposes as

$$\mathbf{U}_X \mathbf{\Sigma}_X^2 \mathbf{U}_X^T = \mathbf{U}_X \mathbf{\Sigma}_X \mathbf{U}_X^T \mathbf{U}_X \mathbf{\Sigma}_X \mathbf{U}_X^T = \mathbf{XX}^T + \lambda_2\mathbf{I}. \tag{23}$$

Let

$$\mathbf{A} = \mathbf{BD}, \mathbf{D} = \mathbf{U}_X \mathbf{\Sigma}^{-1} \mathbf{U}_X^T, \tag{24}$$

Define

$$\begin{aligned} g_1 = \mathbf{BDXY}^T, g_2 = \mathbf{BB}^T, \qquad g_3 = (\mathbf{BDD}^T\mathbf{B}^T)(\mathbf{BDD}^T\mathbf{B}^T)^T \\ g_4 = (\mathbf{BDD}^T\mathbf{B}^T), \qquad g_5 = \log|\det \mathbf{BDD}^T\mathbf{B}^T|, \end{aligned} \tag{25}$$

Then (18) equvalently is

$$\min_{\mathbf{B}} -Tr\{g_1\} + Tr\{g_2\} + \lambda_3 Tr\{g_3 - g_4\} - \lambda_4 g_5. \tag{26}$$

Asumme that $\mathbf{B}$ decomposes as

$$\mathbf{U}_B \mathbf{\Sigma}_B \mathbf{V}_B^T \tag{27}$$

where $\mathbf{\Sigma}_B$ is a diagonal matrix with positive diagonal elements, $\mathbf{U}_B$ is column orthogonal and $\mathbf{V}_B$ is orthogonal square matrix. Moreover, let the following decomposition on $\mathbf{XY}^T$ exists

$$\mathbf{XY}^T = \mathbf{U}_X \mathbf{\Sigma}_{XY} \mathbf{V}_{XY}^T, \text{ and substitute as } \mathbf{U}_B = \mathbf{V}_{XY}, \mathbf{V}_B = \mathbf{U}_X, \tag{28}$$

then

$$Tr\{g_1\} = Tr\{\mathbf{U}_B \mathbf{\Sigma}_B \mathbf{V}_B^T \mathbf{U}_X \mathbf{\Sigma}_X^{-1} \mathbf{U}_X^T \mathbf{XY}^T\} = Tr\{\mathbf{\Sigma}_B \mathbf{\Sigma}_X^{-1} \mathbf{\Sigma}_{XY}\}. \tag{29}$$

The term

$$Tr\{g_2\} = \{\mathbf{BB}^T\} = Tr\{(\mathbf{U}_B \mathbf{\Sigma}_B \mathbf{V}_B^T)(\mathbf{U}_B \mathbf{\Sigma}_B \mathbf{V}_B^T)^T\} = Tr\{\mathbf{\Sigma}_B^2\}, \tag{30}$$

and

$$Tr\{g_4\} = \{\mathbf{BDD}^T\mathbf{B}^T\} = Tr\{\mathbf{\Sigma}_B \mathbf{\Sigma}_X^{-1} \mathbf{\Sigma}_X^{-1} \mathbf{\Sigma}_B\} = Tr\{\mathbf{\Sigma}_X^{-2}\mathbf{\Sigma}_B^2\}. \tag{31}$$

$$Tr\{g_3\} = \{(\mathbf{BDD}^T\mathbf{B}^T)(\mathbf{BDD}^T\mathbf{B}^T)^T\} = Tr\{\mathbf{\Sigma}_X^{-4}\mathbf{\Sigma}_B^4\}, \tag{32}$$

$$g_5 = \log |\det \mathbf{A}\mathbf{A}^T| = \log |\det \mathbf{D}^T\mathbf{B}^T\mathbf{B}\mathbf{D}| = \\ \log |\det \mathbf{U}_X\mathbf{\Sigma}_X^{-1}\mathbf{\Sigma}_B^2\mathbf{\Sigma}_X^{-1}\mathbf{U}_X^T| = \log |\det \mathbf{\Sigma}_X^{-2}\mathbf{\Sigma}_B^2| \tag{33}$$

Finnaly, (18) is reduced to

$$min_{\sigma_B(n)} \sum_{n=1}^{N} \frac{\lambda_3}{\sigma_X^4(n)}\sigma_B^4(n) + \frac{\sigma_X^2(n) - 2\lambda_3}{\sigma_X^2(n)}\sigma_B^2(n) - \frac{\sigma_{XY}(n)}{\sigma_X(n)}\sigma_B(n) - 2\lambda_4 \log\frac{\sigma_X(n)}{\sigma_A(n)}, \tag{34}$$

equalling to zero the first order derivative of the objective (33) w.r.t. $\sigma_B(n)$ and multiplaing by $\sigma_B(n)$ gives

$$4\frac{\lambda_3}{\sigma_X^4(n)}\sigma_B^4(n) + 2\frac{\sigma_X^2(n) - 2\lambda_3}{\sigma_X^2(n)}\sigma_B^2(n) - \frac{\sigma_{XY}(n)}{\sigma_X(n)}\sigma_B(n) - 2\lambda_4 = 0 \tag{35}$$

A closed form solution to (34) exists and depends on the discriminint of the quartic polynomial. Moreover, since $4\frac{\lambda_3}{\sigma_X^4(n)}$ is positive a global minimum to (18) exists if and only if the decompsition (19) exists □

## A.2 THE $\epsilon$-CLOSE CLOSED FORM APPROXIMATION

Consider the equvalent trace form (26) of (18) and note that since $\lambda_2 \geq 0$, $\mathbf{X}\mathbf{X}^T + \lambda_2\mathbf{I}$ is a symetric positive definite matrix whit all eigenvalues non-negative, therfore it decomposes as

$$\mathbf{U}_X\mathbf{\Sigma}_X^2\mathbf{U}_X^T = \mathbf{U}_X\mathbf{\Sigma}_X\mathbf{U}_X^T\mathbf{U}_X\mathbf{\Sigma}_X\mathbf{U}_X^T = \mathbf{X}\mathbf{X}^T + \lambda_2\mathbf{I}. \tag{36}$$

Let the following decomposition exists

$$\mathbf{U}_{U_X XY}\mathbf{\Sigma}_{U_X XY}\mathbf{V}_{U_X XY}^T = \mathbf{U}_X\mathbf{X}\mathbf{Y}^T. \tag{37}$$

Define

$$\mathbf{A} = \mathbf{B}\mathbf{D}, \text{ where } \mathbf{D} = \mathbf{U}_X\mathbf{\Sigma}_X^{-1}\mathbf{U}_X^T \tag{38}$$

Assume that $\mathbf{B}$ decomposes as $\mathbf{U}_B\mathbf{\Sigma}_B\mathbf{V}_B^T = \mathbf{B}$, where $\mathbf{\Sigma}_B$ is a diagonal matrix with positive diagonal elements, $\mathbf{U}_B$ is column orthogonal and $\mathbf{V}_B$ is orthogonal square matrix and let

$$\mathbf{U}_B = (\mathbf{U}_{U_X XY}\mathbf{V}_{U_X XY}^T)^T, \mathbf{V}_B = \mathbf{U}_X, \tag{39}$$

then

$$Tr\{\mathbf{A}\mathbf{X}\mathbf{Y}^T\} = Tr\{\mathbf{B}\mathbf{U}_X\mathbf{\Sigma}_X^{-1}\mathbf{U}_X^T\mathbf{X}\mathbf{Y}^T\} = \\ Tr\{\mathbf{V}_{U_X XY}\mathbf{U}_{U_X XY}^T\mathbf{\Sigma}_B\mathbf{\Sigma}_X^{-1}\mathbf{U}_{U_X XY}\mathbf{U}_{U_X XY}\mathbf{\Sigma}_{V_X XY}^T\}. \tag{40}$$

Consider the decomposition $\mathbf{U}_B\mathbf{\Sigma}_B\mathbf{V}_B^T$ of $\mathbf{B}$, use Mirsky (1959) and Neumann (1937) and note that

$$\min_{\mathbf{\Sigma}_B} \max_{\mathbf{U}_B, \mathbf{V}_B} Tr\{\mathbf{U}_B\mathbf{\Sigma}_B\mathbf{V}_B^T\mathbf{U}_X\mathbf{\Sigma}_X^{-1}\mathbf{U}_X^T\mathbf{X}\mathbf{Y}^T\} \leq \min_{\mathbf{\Sigma}_B} Tr\{\mathbf{\Sigma}_B\mathbf{\Sigma}_X^{-1}\mathbf{\Sigma}_\Gamma\}, \tag{41}$$

where $\mathbf{\Sigma}_\Gamma$ is a diagonal matrix, having diagonal elements $\Sigma_\Gamma(n,n) = \sigma_\Gamma(n) = T(n,n), \forall n \in \mathcal{N}$ and $\mathbf{T} = \mathbf{U}_{U_X XY}\mathbf{\Sigma}_{U_X XY}\mathbf{V}_{U_X XY}^T$.

Note that the term $Tr\{(\mathbf{A}\mathbf{X})(\mathbf{A}\mathbf{X})^T\} = Tr\{\mathbf{B}\mathbf{B}^T\} = Tr\{\mathbf{\Sigma}_B^2\}$ and as in the *Apendix subsection A.1* $Tr\{\mathbf{A}\mathbf{A}^T\} = Tr\{\mathbf{B}\mathbf{D}\mathbf{D}^T\mathbf{B}^T\} = Tr\{\mathbf{\Sigma}_B^2\mathbf{\Sigma}_X^{-2}\}$, $Tr\{(\mathbf{A}\mathbf{A}^T)(\mathbf{A}\mathbf{A}^T)^T\} = Tr\{(\mathbf{B}\mathbf{D}\mathbf{D}^T\mathbf{B}^T)(\mathbf{B}\mathbf{D}\mathbf{D}^T\mathbf{B}^T)\} = Tr\{\mathbf{\Sigma}_B^4\mathbf{\Sigma}_X^{-4}\}$ and $\log |\det \mathbf{A}\mathbf{A}^T| = \log |\det \mathbf{D}^T\mathbf{B}^T\mathbf{B}\mathbf{D}| = \log |\det \mathbf{U}_X\mathbf{\Sigma}_X^{-1}\mathbf{\Sigma}_B^2\mathbf{\Sigma}_X^{-1}\mathbf{U}_X^T| = \log |\det \mathbf{\Sigma}_X^{-2}\mathbf{\Sigma}_B^2|$.

Finally, the aproximation of (18) using the bound (41) is reduced to

$$min_{\sigma_B(n)} \sum_{n=1}^{N} \frac{\lambda_3}{\sigma_X^4(n)}\sigma_B^4(n) + \frac{\sigma_X^2(n) - 2\lambda_3}{\sigma_X^2(n)}\sigma_B^2(n) - \frac{\sigma_\Gamma(n)}{\sigma_X(n)}\sigma_B(n) - 2\lambda_4 \log\frac{\sigma_X(n)}{\sigma_A(n)}, \tag{42}$$

equalling to zero the first order derivative of the objective (42) w.r.t. $\sigma_B(n)$ and multiplaing by $\sigma_B(n)$ gives

$$4\frac{\lambda_3}{\sigma_X^4(n)}\sigma_B^4(n) + 2\frac{\sigma_X^2(n) - 2\lambda_3}{\sigma_X^2(n)}\sigma_B^2(n) - \frac{\sigma_\Gamma(n)}{\sigma_X(n)}\sigma_B(n) - 2\lambda_4 = 0. \tag{43}$$

A closed form solution to (43) exists and depends on the discriminint of the quartic polynomial. Moreover, since $4 \frac{\lambda_3}{\sigma_X^4(n)}$ is positive a global minimum to (42) exists. Therfore, having the decomposition $\mathbf{U}_B \mathbf{\Sigma}_B \mathbf{V}_B^T = \mathbf{B}$, the substitutions $\mathbf{U}_B = (\mathbf{U}_{U_X XY} \mathbf{V}_{U_X XY}^T)^T$ and $\mathbf{V}_B = \mathbf{U}_X$ with the solution of (42) gives the $\epsilon$-close closed form approximative solution to problem (18) as

$$\mathbf{A} = \mathbf{V}_{U_X XY} \mathbf{U}_{U_X XY}^T \mathbf{\Sigma}_B \mathbf{\Sigma}_X^{-1} \mathbf{U}_X^T, \tag{44}$$

where the bound (41) implies that the $\epsilon$-close closed form approximative solution is a lower bound to the solution of (18) □

## APPENDIX B.

Let $\mathbf{y}_{c1,k1} = \mathbf{y}_{c1,k1}^+ + \mathbf{y}_{c1,k1}^-, \mathbf{y}_{c1,k1}^+ \in \Re_+^M$ and $\mathbf{y}_{c1,k1}^- \in \Re_-^M$. Consider the measure $D_{\ell_1}^P(\mathbf{X})$

$$D_{\ell_1}^P(\mathbf{X}) = \sum_{\substack{c1,c2 \\ c1 \neq c2}} \sum_{k1,k2} \|\mathbf{y}_{c1,k1}^+ \odot \mathbf{y}_{c2,k2}^+\|_1 + \sum_{\substack{c1,c2 \\ c1 \neq c2}} \sum_{k1,k2} \|\mathbf{y}_{c1,k1}^- \odot \mathbf{y}_{c2,k2}^-\|_1 =$$

$$\sum_{\substack{c1,c2 \\ c1 \neq c2}} \sum_{k1,k2} |\mathbf{y}_{c1,k1}^+|^T |\mathbf{y}_{c2,k2}^+| + \sum_{\substack{c1,c2 \\ c1 \neq c2}} \sum_{k1,k2} |\mathbf{y}_{c1,k1}^-|^T |\mathbf{y}_{c2,k2}^-|. \tag{45}$$

Let $\mathbf{A}$ and $\mathbf{Y}_{\setminus\{c1,k1\}}$ be given then problem $P_{DP}$ has only one varible $\mathbf{y}_{c1,k1}$. Conseqently in (45), $\mathbf{y}_{c1,k1}$ is releted with only a part of the transform representations in $D_{\ell_1}^P(\mathbf{X})$, the rest are constants for the reduced problem $P_{DP}$, in particulary we have

$$|\mathbf{y}_{c1,k1}^+|^T \sum_{\substack{c2 \\ c2 \neq c1}} \sum_{k2} |\mathbf{y}_{c2,k2}^+| + |\mathbf{y}_{c1,k1}^-|^T \sum_{\substack{c2 \\ c2 \neq c1}} \sum_{k2} |\mathbf{y}_{c2,k2}^-| = |\mathbf{y}_{c1,k1}^+|^T \mathbf{d}_c^+ + |\mathbf{y}_{c1,k1}^-|^T \mathbf{d}_c^-, \tag{46}$$

where $\mathbf{d}_c^+ = \sum_{\substack{c2, \\ c2 \neq c1}} \sum_{k2} |\mathbf{y}_{c2,k2}^+|$, $\mathbf{d}_c^- = \sum_{\substack{c2 \\ c2 \neq c1}} \sum_{k2} |\mathbf{y}_{c2,k2}^-|$ and we abuse notation by denoting $|\mathbf{y}_{c1,k1}|$ as the vector whose elements are the absolute values of the elements in $\mathbf{y}_{c1,k1}$.

Note that

$$S_{\ell_2}^P(\mathbf{X}) = \sum_{\substack{c1,c2 \\ c1 \neq c2}} \sum_{k1,k2} \|\mathbf{y}_{c1,k1} \odot \mathbf{y}_{c2,k2}\|_2^2 =$$

$$\sum_{\substack{c1,c2 \\ c1 \neq c2}} \sum_{k1,k2} (\mathbf{y}_{c1,k1} \odot \mathbf{y}_{c1,k1})^T (\mathbf{y}_{c2,k2} \odot \mathbf{y}_{c2,k2}). \tag{47}$$

simmilary as in (46) we have

$$\sum_{\substack{c2 \\ c2 \neq c1}} \sum_{k2} (\mathbf{y}_{c1,k1} \odot \mathbf{y}_{c1,k1})^T (\mathbf{y}_{c2,k2} \odot \mathbf{y}_{c2,k2}) =$$

$$(\mathbf{y}_{c1,k1} \odot \mathbf{y}_{c1,k1})^T (\sum_{\substack{c2 \\ c2 \neq c1}} \sum_{k2} \mathbf{y}_{c2,k2} \odot \mathbf{y}_{c2,k2}) = (\mathbf{y}_{c1,k1} \odot \mathbf{y}_{c1,k1})^T \mathbf{s}_c, \tag{48}$$

where $\mathbf{s}_c = \sum_{\substack{c2 \\ c2 \neq c1}} \sum_{k2} \mathbf{y}_{c2,k2} \odot \mathbf{y}_{c2,k2}$. Denote $\mathbf{q}_{c1,k1} = \mathbf{A}\mathbf{x}_{c1,k1}$ and consider the problem

$$\min_{\mathbf{y}_{c1,k1}} \|\mathbf{q}_{c1,k1} - \mathbf{y}_{c1,k1}\|_2^2 +$$
$$\lambda_0((\mathbf{y}_{c1,k1}^+)^T \mathbf{d}_c^+ + (\mathbf{y}_{c1,k1}^-)^T \mathbf{d}_c^- + (\mathbf{y}_{c1,k1} \odot \mathbf{y}_{c1,k1})^T \mathbf{s}_c) + \lambda_1 \|\mathbf{y}_{c1,k1}\|_1, \tag{49}$$

by taking the first order derivative w.r.t. $\mathbf{y}_{c1,k1}$ we have that

$$(\mathbf{y}_{c1,k1} - \mathbf{q}_{c1,k1}) + \lambda_0 (\text{sign}(\mathbf{y}_{c1,k1}^+) \odot \mathbf{d}_c^+ + \text{sign}(\mathbf{y}_{c1,k1}^-) \odot \mathbf{d}_c^- + \mathbf{y}_{c1,k1} \odot \mathbf{s}_c)$$
$$\lambda_1 \text{sign}(\mathbf{y}_{c1,k1}) = \mathbf{0}, \tag{50}$$

take sign magnitude decomposition of $\mathbf{y}_{c1,k1} = \text{sign}(\mathbf{y}_{c1,k1}) \odot |\mathbf{y}_{c1,k1}|$ then we have

$$\text{sign}(\mathbf{y}_{c1,k1}) \odot |\mathbf{y}_{c1,k1}| \odot (\mathbf{1} + 2\lambda_0 \mathbf{s}_c) - \text{sign}(\mathbf{q}_{c1,k1}) \odot |\mathbf{q}_{c1,k1}| +$$
$$\lambda_0 (\text{sign}(\mathbf{y}_{c1,k1}^+) \odot \mathbf{d}_c^+ + \text{sign}(\mathbf{y}_{c1,k1}^-) \odot \mathbf{d}_c^-) + \lambda_1 \text{sign}(\mathbf{y}_{c1,k1}) = \mathbf{0}. \tag{51}$$

$$\mathbf{X} \xrightarrow{\mathcal{T}^{\mathcal{P}}} \mathbf{Y}$$

$$\downarrow \qquad\qquad\qquad \downarrow$$

$$\log \frac{D_{\ell_1,c}^{\mathcal{B}^N}(\mathbf{X})}{D_{\ell_1}^{\mathcal{B}^N}(\mathbf{X})+\epsilon} \qquad \log \frac{D_{\ell_1,c}^{\mathcal{P}}(\mathbf{X})}{D_{\ell_1}^{\mathcal{P}}(\mathbf{X})+\epsilon} = \log \frac{D_{\ell_1,c}^{\mathcal{B}^M}(\mathbf{Y})}{D_{\ell_1}^{\mathcal{B}^M}(\mathbf{Y})+\epsilon}$$

$$\downarrow \qquad\qquad\qquad\qquad \downarrow$$

$$\mathcal{I}^o \qquad\qquad\qquad \mathcal{I}^t$$

Figure 7: The relation for the definition of the discrimination power in the original and the transform domain under the base models $\mathcal{B}^N$ and $\mathcal{B}^M$.

Let the sign of $\mathbf{y}_{c1,k1}$, *i.e.* $\mathrm{sign}(\mathbf{y}_{c1,k1})$ be equal to the sign of $\mathrm{sign}(\mathbf{q}_{c1,k1})$, and Hadamar multiply from the left side by $\mathrm{sign}(\mathbf{q}_{c1,k1})$ then we have

$$|\mathbf{y}_{c1,k1}| \odot (\mathbf{1} + 2\lambda_0 \mathbf{s}_c) - |\mathbf{q}_{c1,k1}| + \lambda_0(\mathrm{sign}(\mathbf{q}_{c1,k1}) \odot \mathrm{sign}(\mathbf{q}_{c1,k1}^+) \odot \mathbf{d}_c^+ +$$

$$\mathrm{sign}(\mathbf{q}_{c1,k1}) \odot \mathrm{sign}(\mathbf{q}_{c1,k1}^-) \odot \mathbf{d}_c^-) + \lambda_1 \mathbf{1} = \mathbf{0}, \tag{52}$$

note that $\mathrm{sign}(\mathbf{q}_{c1,k1}) \odot \mathrm{sign}(\mathbf{q}_{c1,k1}^+) = \mathrm{sign}(\mathbf{q}_{c1,k1}^+)$ and that $\mathrm{sign}(\mathbf{q}_{c1,k1}) \odot \mathrm{sign}(\mathbf{q}_{c1,k1}^-) = \mathrm{sign}(-\mathbf{q}_{c1,k1}^-)$, theretofore we have

$$|\mathbf{y}_{c1,k1}| \odot (\mathbf{1} + 2\lambda_0 \mathbf{s}_c) = |\mathbf{q}_{c1,k1}| - \lambda_0(\mathrm{sign}(\mathbf{q}_{c1,k1}^+) \odot \mathbf{d}_c^+ + \mathrm{sign}(-\mathbf{q}_{c1,k1}^-) \odot \mathbf{d}_c^-) -$$

$$\lambda_1 \mathbf{1}, \tag{53}$$

since the magnitude might be only positive we have that $|\mathbf{y}_{c1,k1}| \odot (\mathbf{1} + 2\lambda_0 \mathbf{s}_c) = \max(|\mathbf{q}_{c1,k1}| - \lambda_0(\mathrm{sign}(\mathbf{q}_{c1,k1}^+) \odot \mathbf{d}_c^+ + \mathrm{sign}(-\mathbf{q}_{c1,k1}^-) \odot \mathbf{d}_c^-) - \lambda_1 \mathbf{1}, \mathbf{0})$. Denote $\mathbf{g}_c = (\mathrm{sign}(\mathbf{q}_{c1,k1}^+) \odot \mathbf{d}_c^+ + \mathrm{sign}(-\mathbf{q}_{c1,k1}^-) \odot \mathbf{d}_c^-)$ then the closed form solution to (49) is:

$$\mathbf{y}_{c1,k1} = \mathrm{sign}(\mathbf{A}\mathbf{x}_{c1,k1}) \odot \max(|\mathbf{A}\mathbf{x}_{c1,k1}| - \lambda_0 \mathbf{g}_c - \lambda_1 \mathbf{1}, \mathbf{0}) \oslash (\mathbf{1} + \lambda_0 \mathbf{s}_c), \tag{54}$$

which completes the proof $\square$

## APPENDIX C. SENSITIVITY ANALYSIS AND INTERPRETATIONS

The similarity concentration measure provides possibility to measure the discriminative properties, their deviation, increase (or decrease) and the corresponding relations between different non-linear transform models across one domain or different domains, thereby quantifying their quality w.r.t. the discriminative properties.

### C.1 SENSITIVITY ANALYSIS W.R.T. THE SIMILARITY CONCENTRATION MEASURES

An illustration about the definition of discriminative power given by a diagram is shown in Figure 7.

To measure the ability for an increase in discriminative properties by a non-linear transform[9] we first have to define a notion for the discriminative properties on a data set under different non-linear transform models. Therefore, first we introduce the "special" *base models* and then analyze the properties of the similarity concentration measures under the change in model parameter and the relation between the *base model* and the proposed non-linear transform model defined by a parameter set $\mathcal{P} = \{\mathbf{A} \in \Re^{M \times N}, \boldsymbol{\tau} \in \Re^M\}$.

Any data set $\mathbf{X}$ in the original domain might have a transform model with parameters $\mathcal{B}^N = \{\mathbf{A}_o \in \Re^{N \times N}, \boldsymbol{\tau} = \mathbf{0} \in \Re_+^N\}$, if $\mathbf{A}_o = \mathbf{I} \in \mathcal{D}_+^N$ we refer to it as a *base original model*. Similarly as in the original domain, any data set $\mathbf{Y}$ in the transform domain might have a transform model with parameters $\mathcal{B}^M = \{\mathbf{A}_t \in \Re^{M \times M}, \boldsymbol{\tau} = \mathbf{0} \in \Re_+^M\}$, if $\mathbf{A}_t = \mathbf{I} \in \mathcal{D}_+^M$ we refer to it as a *base transform model*. Any base model, defined ether in the original domain $\mathcal{B}^N$ or in the transform domain $\mathcal{B}^M$, has domain equal to the co-domain, since $\mathbf{x}_{c,k} = \mathcal{T}^{\mathcal{B}^N}(\mathbf{x}_{c,k})$ and $\mathbf{y}_{c,k} = \mathcal{T}^{\mathcal{B}^M}(\mathbf{y}_{c,k})$ holds trivially, for the respective sets of parameters $\mathcal{B}^N = \{\mathbf{A}_o = \mathbf{I} \in \mathcal{D}_+^N, \boldsymbol{\tau} = \mathbf{0} \in \Re_+^N\}$ and $\mathcal{B}^M = \{\mathbf{A}_t = \mathbf{I} \in \mathcal{D}_+^M, \boldsymbol{\tau} = \mathbf{0} \in \Re_+^M\}$. This is illustrated with a diagram shown in Figure 8.

---

[9]Instead of using the term non-liner transform model as defined by (1) or (2) for short we just use the term model.

$$
\begin{array}{ccc}
\text{original domain} & & \text{transform domain} \\
\mathbf{X} & \xrightarrow{\mathcal{T}^{\mathcal{P}}} & \mathbf{Y} \\
\updownarrow \mathcal{T}^{\mathcal{B}^N} & & \updownarrow \mathcal{T}^{\mathcal{B}^M} \\
\mathbf{X} & & \mathbf{Y} \\
\text{transform domain} & & \text{transform domain}
\end{array}
$$

Figure 8: The original and the transform domains under a non-linear transforms with a set of parameters $\mathcal{B}^N$, $\mathcal{P}$ and $\mathcal{B}^M$, note that for $\mathcal{T}^{\mathcal{B}^N}$ and $\mathcal{T}^{\mathcal{B}^M}$ the original and the transform domains are the same.

A base original model provides a possibility to compare it with any other non-linear transform model with parameters $\mathcal{P} = \{\mathbf{A} \in \Re^{M \times N}, \boldsymbol{\tau} \in \Re_+^M\}$. Additionally, note that for $\mathcal{B}^M = \{\mathbf{A}_t = \mathbf{I} \in \mathcal{D}^M, \boldsymbol{\tau} = \mathbf{0} \in \Re^M\}$ we have that $D_{\ell_1,c}^{\mathcal{B}^M}(\mathbf{Y}) = D_{\ell_1,c}^{\mathcal{P}}(\mathbf{X})$ and that $D_{\ell_1}^{\mathcal{B}^M}(\mathbf{Y}) = D_{\ell_1}^{\mathcal{P}}(\mathbf{X})$. It implies that the similarity concentrations can be analyzed as a function in the original domain under model $\mathcal{P}$ or in the transform domain under model $\mathcal{B}^M$. The main relations considering the preservation of change in the similarity concentration between two models, defined not necessary in the same domain are stated by **Lemma 1**.

**Lemma 1**: *The non-linear transform model* (2) *totally preserves the information in the change for the similarity concentration for a data set* $\mathbf{X}$ *w.r.t. a small change in the parameters of the models* $\mathcal{B}^N$, $\mathcal{B}^M$ *and* $\mathcal{P}$ *if* $\|\boldsymbol{\delta}_o\|_* = 0$ *and* $\|\boldsymbol{\delta}_t\|_* = 0$ *as*

$$
(R1): \mathbf{A}\left(\frac{\partial D_{\ell_1,c}^{\mathcal{B}^N}(\mathbf{X})}{\partial \mathbf{A}_o}\Big|_{\mathbf{A}_o=\mathbf{I}} + \frac{\partial D_{\ell_1}^{\mathcal{B}^N}(\mathbf{X})}{\partial \mathbf{A}_o}\Big|_{\mathbf{A}_o=\mathbf{I}}\right) = \frac{\partial D_{\ell_1,c}^{\mathcal{P}}(\mathbf{X})}{\partial \mathbf{A}} + \frac{\partial D_{\ell_1}^{\mathcal{P}}(\mathbf{X})}{\partial \mathbf{A}} + \boldsymbol{\delta}_o
$$

$$
(R2): \mathbf{A}\left(\frac{\partial D_{\ell_1,c}^{\mathcal{P}}(\mathbf{X})}{\partial \mathbf{A}} + \frac{\partial D_{\ell_1}^{\mathcal{P}}(\mathbf{X})}{\partial \mathbf{A}}\right)^T = \frac{\partial D_{\ell_1,c}^{\mathcal{B}^M}(\mathbf{Y})}{\partial \mathbf{A}_t}\Big|_{\mathbf{A}_t=\mathbf{I}} + \frac{\partial D_{\ell_1}^{\mathcal{B}^M}(\mathbf{Y})}{\partial \mathbf{A}_t}\Big|_{\mathbf{A}_t=\mathbf{I}} + \boldsymbol{\delta}_t \quad (55)
$$

$$
(R3): \mathbf{A}\boldsymbol{\delta}_o^T = \frac{\partial D_{\ell_1,c}^{\mathcal{B}^M}(\mathbf{Z})}{\partial \mathbf{A}_t}\Big|_{\mathbf{A}_t=\mathbf{I}} + \frac{\partial D_{\ell_1}^{\mathcal{B}^M}(\mathbf{Z})}{\partial \mathbf{A}_t}\Big|_{\mathbf{A}_t=\mathbf{I}} + \boldsymbol{\delta}_t^T,
$$

*where*

$$
\frac{\partial D_{\ell_1,c}^{\mathcal{P}}(\mathbf{X})}{\partial \mathbf{A}} + \frac{\partial D_{\ell_1}^{\mathcal{P}}(\mathbf{X})}{\partial \mathbf{A}} = \sum_{c,c1}\sum_{\substack{k,k1 \\ k \neq k1}} \mathbf{y}_{c1,k1}\mathbf{x}_{c,k}^T + \mathbf{y}_{c,k}\mathbf{x}_{c1,k1}^T
$$

$$
\frac{\partial D_{\ell_1,c}^{\mathcal{B}^M}(\mathbf{Y})}{\partial \mathbf{A}_t}\Big|_{\mathbf{A}_t=\mathbf{I}} + \frac{\partial D_{\ell_1}^{\mathcal{B}^M}(\mathbf{Y})}{\partial \mathbf{A}_t}\Big|_{\mathbf{A}_t=\mathbf{I}} = \sum_{c,c1}\sum_{\substack{k,k1 \\ k \neq k1}} \mathbf{y}_{c1,k1}\mathbf{y}_{c,k}^T + \mathbf{y}_{c,k}\mathbf{y}_{c1,k1}^T
$$

$$
\frac{\partial D_{\ell_1,c}^{\mathcal{B}^M}(\mathbf{Z})}{\partial \mathbf{A}_t}\Big|_{\mathbf{A}_t=\mathbf{I}} + \frac{\partial D_{\ell_1}^{\mathcal{B}^M}(\mathbf{Z})}{\partial \mathbf{A}_t}\Big|_{\mathbf{A}_t=\mathbf{I}} = \sum_{c,c1}\sum_{\substack{k,k1 \\ k \neq k1}} \mathbf{z}_{c1,k1}\mathbf{z}_{c,k}^T + \mathbf{z}_{c,k}\mathbf{z}_{c1,k1}^T
$$

$$
\frac{\partial D_{\ell_1,c}^{\mathcal{B}^N}(\mathbf{X})}{\partial \mathbf{A}_o}\Big|_{\mathbf{A}_o=\mathbf{I}} = \sum_{c}\sum_{k,k1} \mathbf{x}_{c,k}\mathbf{x}_{c,k1}^T + \mathbf{x}_{c,k1}\mathbf{x}_{c,k}^T
$$

$$
\frac{\partial D_{\ell_1}^{\mathcal{B}^N}(\mathbf{X})}{\partial \mathbf{A}_o}\Big|_{\mathbf{A}_o=\mathbf{I}} = \sum_{c,c1,c \neq c1}\sum_{k,k1} \mathbf{x}_{c,k}\mathbf{x}_{c1,k1}^T + \mathbf{x}_{c1,k1}\mathbf{x}_{c,k}^T \quad (56)
$$

$$
\boldsymbol{\delta}_o = \sum_{c,c1}\sum_{\substack{k,k1 \\ k \neq k1}} \mathbf{z}_{c1,k1}\mathbf{x}_{c,k}^T + \mathbf{z}_{c,k}\mathbf{x}_{c1,k1}^T
$$

$$
\boldsymbol{\delta}_t = \sum_{c,c1}\sum_{\substack{k,k1 \\ k \neq k1}} \mathbf{z}_{c1,k1}\mathbf{y}_{c,k}^T + \mathbf{z}_{c,k}\mathbf{y}_{c1,k1}^T
$$

*The proof is given in Appendix C.2.*

The terms $\mathbf{z}_{c,k}$ represent the non-linear transform error vectors that appear in the model $\mathbf{A}\mathbf{x}_{c,k} = \mathbf{y}_{c,k} + \mathbf{z}_{c,k}$ as a result of applying an element-wise non-liner operation $\mathcal{H}_\tau$ to $\mathbf{A}\mathbf{x}_{c,k}$, *i.e.*, $\mathbf{y}_{c,k} = \mathcal{H}_\tau(\mathbf{A}\mathbf{x}_{c,k})$. As an example in the sparsifying transform model $\mathbf{z}_{c,k}$ is the "loss of information", that is the information about the values of the elements in $\mathbf{A}\mathbf{x}_{c,k}$ that are discarded. The terms $\boldsymbol{\delta}_o$ and $\boldsymbol{\delta}_t$ correlate the errors $\mathbf{z}_{c,k}$ with the original data $\mathbf{x}_{c1,k1}$ and transform data $\mathbf{y}_{c1,k1}$, respectively. Note that if there is no loss of information (in the earlier example it means that there is no thresholding and just a simple linear transform model is used) then $\boldsymbol{\delta}_o = \mathbf{0}$ and $\boldsymbol{\delta}_t = \mathbf{0}$. Moreover, $\boldsymbol{\delta}_o$ and $\boldsymbol{\delta}_t$ bear important information about the discriminative properties in the transform domain.

The terms $\frac{\partial D_{\ell_1,c}^{\mathcal{P}}(\mathbf{X})}{\partial \mathbf{A}}$ and $\frac{\partial D_{\ell_1}^{\mathcal{P}}(\mathbf{X})}{\partial \mathbf{A}}$ represent the change of the similarity concentrations under infinitesimally small change of the parameter $\mathbf{A}$ from the model $\mathcal{P}$. The terms $\frac{\partial D_{\ell_1,c}^{\mathcal{B}^N}(\mathbf{X})}{\partial \mathbf{A}_o}|_{\mathbf{A}_o=\mathbf{I}}$ and $\frac{\partial D_{\ell_1}^{\mathcal{B}^N}(\mathbf{X})}{\partial \mathbf{A}_o}|_{\mathbf{A}_o=\mathbf{I}}$ have dual interpretation. Assuming metric $\mathbf{A}_o = \mathbf{I}$, then the first one is considered as a change of the similarity concentrations under infinitesimally small change of the space metric, or equivalently under small metric perturbation. Conversely, assuming the data samples are distributed under a Gaussian distribution with parameters identity covariance matrix and zero mean, *i.e.* $\mathbf{x}_{c,k} \sim \mathcal{N}(\boldsymbol{\mu} = \mathbf{0}, \boldsymbol{\Sigma} = \mathbf{I})$, then $\frac{\partial D_{\ell_1,c}^{\mathcal{B}^N}(\mathbf{X})}{\partial \mathbf{A}_o}|_{\mathbf{A}_o=\mathbf{I}}$ and $\frac{\partial D_{\ell_1}^{\mathcal{B}^N}(\mathbf{X})}{\partial \mathbf{A}_o}|_{\mathbf{A}_o=\mathbf{I}}$ represent the change of the similarity concentrations under small change in the assumption away from a Gaussian distribution.

Equation $(R1)$ relates the base transform for the original domain $\mathcal{B}^N$ with any arbitrary transform defined in the original domain $\mathcal{P}$. The relation $(R2)$ is a result about the preservation of change in the similarity concentration between two models $\mathcal{B}^M$ and $\mathcal{P}$ defined on two different domains. Whereas $(R3)$ gives the preservation of change in the similarity concentration between the error in the transform domain.

The next result highlights the relation between: *the linear projection (by the linear map $\mathbf{A}$ that appears in the model $\mathcal{P}$) of the change in the similarity concentration under the model $\mathcal{B}^N$ in the original domain and the change of the similarity concentration under the model $\mathcal{B}^M$ in the transform domain.*

This relation exists independently for $\frac{\partial D_{\ell_1,c}^{\mathcal{B}^N}(\mathbf{X})}{\partial \mathbf{A}_o}|_{\mathbf{A}_o=\mathbf{I}}$ and $\frac{\partial D_{\ell_1}^{\mathcal{B}^N}(\mathbf{X})}{\partial \mathbf{A}_o}|_{\mathbf{A}_o=\mathbf{I}}$, nevertheless, we will define the summarized and the independent versions. Therefore, first we define $\frac{\partial \mathcal{J}_{\ell_1}(\mathbf{X})}{\partial \mathbf{A}_o}|_{\mathbf{A}_o=\mathbf{I}} = \frac{\partial D_{\ell_1,c}^{\mathcal{B}^N}(\mathbf{X})}{\partial \mathbf{A}_o}|_{\mathbf{A}_o=\mathbf{I}} + \frac{\partial D_{\ell_1}^{\mathcal{B}^N}(\mathbf{X})}{\partial \mathbf{A}_o}|_{\mathbf{A}_o=\mathbf{I}}$ and $\frac{\partial \mathcal{J}_{\ell_1}(\mathbf{Y})}{\partial \mathbf{A}_t}|_{\mathbf{A}_t=\mathbf{I}} = \frac{\partial D_{\ell_1,c}^{\mathcal{B}^N}(\mathbf{Y})}{\partial \mathbf{A}_t}|_{\mathbf{A}_t=\mathbf{I}} + \frac{\partial D_{\ell_1}^{\mathcal{B}^N}(\mathbf{Y})}{\partial \mathbf{A}_t}|_{\mathbf{A}_t=\mathbf{I}}$ and rewrite $(R1)$ as $\mathbf{A}\frac{\partial \mathcal{J}(\mathbf{X})}{\partial \mathbf{A}_o}|_{\mathbf{A}_o=\mathbf{I}} - \boldsymbol{\delta}_o = \frac{\partial D_{\ell_1,c}^{\mathcal{P}}(\mathbf{X})}{\partial \mathbf{A}} + \frac{\partial D_{\ell_1}^{\mathcal{P}}(\mathbf{X})}{\partial \mathbf{A}}$, then replace $\frac{\partial D_{\ell_1,c}^{\mathcal{P}}(\mathbf{X})}{\partial \mathbf{A}} + \frac{\partial D_{\ell_1}^{\mathcal{P}}(\mathbf{X})}{\partial \mathbf{A}}$ in $(R2)$ by the same term in $(R1)$, use $(R3)$, reorder and we have the following result.

***Lemma 2***: *For fixed $\tau$ any non-linear transform model* (2) *preservers the information in the change of similarity concentrations w.r.t. a small change in $\mathbf{A}$ by*

$$(R4): \mathbf{A}\frac{\partial \mathcal{J}_{\ell_1}(\mathbf{X})}{\partial \mathbf{A}_o}|_{\mathbf{A}_o=\mathbf{I}}\mathbf{A}^T = \frac{\partial \mathcal{J}_{\ell_1}(\mathbf{V})}{\partial \mathbf{A}_t}|_{\mathbf{A}_t=\mathbf{I}} = \frac{\partial \mathcal{J}_{\ell_1}(\mathbf{Y})}{\partial \mathbf{A}_t}|_{\mathbf{A}_t=\mathbf{I}} + \boldsymbol{\xi}_c + \boldsymbol{\xi},$$

$$(R5): \mathbf{A}\frac{\partial D_{\ell_1,c}(\mathbf{X})}{\partial \mathbf{A}_o}|_{\mathbf{A}_o=\mathbf{I}}\mathbf{A}^T = \frac{\partial D_{\ell_1}(\mathbf{V})}{\partial \mathbf{A}_t}|_{\mathbf{A}_t=\mathbf{I}} = \frac{\partial D_{\ell_1,c}(\mathbf{Y})}{\partial \mathbf{A}_t}|_{\mathbf{A}_t=\mathbf{I}} + \boldsymbol{\xi}_c, \qquad (57)$$

$$(R6): \mathbf{A}\frac{\partial D_{\ell_1}(\mathbf{X})}{\partial \mathbf{A}_o}|_{\mathbf{A}_o=\mathbf{I}}\mathbf{A}^T = \frac{\partial D_{\ell_1}(\mathbf{V})}{\partial \mathbf{A}_t}|_{\mathbf{A}_t=\mathbf{I}} = \frac{\partial D_{\ell_1}(\mathbf{Y})}{\partial \mathbf{A}_t}|_{\mathbf{A}_t=\mathbf{I}} + \boldsymbol{\xi},$$

*where*

$$\mathbf{V} = \mathbf{A}\mathbf{X}$$

$$\boldsymbol{\xi}_c + \boldsymbol{\xi} = \frac{\partial \mathcal{J}_{\ell_1}(\mathbf{Z})}{\partial \mathbf{A}_t}|_{\mathbf{A}_t=\mathbf{I}} + \frac{\partial \mathcal{J}_{\ell_1}(\mathbf{Y};\mathbf{Z})}{\partial \mathbf{A}_t}|_{\mathbf{A}_t=\mathbf{I}}$$

$$\frac{\partial \mathcal{J}_{\ell_1}(\mathbf{Z})}{\partial \mathbf{A}_t}|_{\mathbf{A}_t=\mathbf{I}} = \frac{\partial D_{\ell_1,c}^{\mathcal{B}^N}(\mathbf{Z})}{\partial \mathbf{A}_t}|_{\mathbf{A}_t=\mathbf{I}} + \frac{\partial D_{\ell_1}^{\mathcal{B}^N}(\mathbf{Z})}{\partial \mathbf{A}_t}|_{\mathbf{A}_t=\mathbf{I}} \qquad (58)$$

$$\frac{\partial \mathcal{J}_{\ell_1}(\mathbf{Y};\mathbf{Z})}{\partial \mathbf{A}_t}|_{\mathbf{A}_t=\mathbf{I}} = \boldsymbol{\delta}_t + \boldsymbol{\delta}_t^T$$

The expressions $(R4)$ actually relate the metric in the original domain under the model $\mathcal{B}^N$ to the induced metric in the transform domain for the model $\mathcal{B}^M$, with induction done by the model $\mathcal{P}$ with parameter set $\{\mathbf{A} \in \Re^{M \times N}, \boldsymbol{\tau} \in \Re_+^M\}$. Moreover, the model $\mathcal{P}$ might describe a transform domain with a non-smooth manifold. Since the manifolds of the original and the transform domain under the models $\mathcal{B}^N$ and $\mathcal{B}^M$ are smooth the analysis of their relations reveals insights about the relation between the manifolds under the models $\mathcal{B}^N$ and $\mathcal{P}$. The terms $\frac{\partial \mathcal{J}_{\ell_1}(\mathbf{Z})}{\partial \mathbf{A}_t}|_{\mathbf{A}_t=\mathbf{I}}$ and $\frac{\partial \mathcal{J}_{\ell_1}(\mathbf{Y};\mathbf{Z})}{\partial \mathbf{A}_t}|_{\mathbf{A}_t=\mathbf{I}}$ carry out the information about the breaks and the discontinuities of the regularity and smoothness in the manifold induced by the model $\mathcal{P}$. Also we note that $\frac{\partial^n \mathcal{J}_{\ell_1}(\mathbf{X})}{\partial^n \mathbf{A}_o}|_{\mathbf{A}_o=\mathbf{I}} = 4\frac{\partial^{n-1} \mathcal{J}_{\ell_1}(\mathbf{X})}{\partial^{n-1} \mathbf{A}_o}|_{\mathbf{A}_o=\mathbf{I}}$ and $\frac{\partial^n \mathcal{J}_{\ell_1}(\mathbf{Y})}{\partial^n \mathbf{A}_t}|_{\mathbf{A}_t=\mathbf{I}} = 4\frac{\partial^{n-1} \mathcal{J}_{\ell_1}(\mathbf{Y})}{\partial^{n-1} \mathbf{A}_t}|_{\mathbf{A}_t=\mathbf{I}}$ therefore, $\frac{\partial \mathcal{J}_{\ell_1}(\mathbf{X})}{\partial \mathbf{A}_o}|_{\mathbf{A}_o=\mathbf{I}} = \frac{1}{4}\frac{\partial^2 \mathcal{J}_{\ell_1}(\mathbf{X})}{\partial^2 \mathbf{A}_o}|_{\mathbf{A}_o=\mathbf{I}}$ and $\frac{\partial \mathcal{J}_{\ell_1}(\mathbf{Y})}{\partial \mathbf{A}_t}|_{\mathbf{A}_t=\mathbf{I}} = \frac{1}{4}\frac{\partial^2 \mathcal{J}_{\ell_1}(\mathbf{Y})}{\partial^2 \mathbf{A}_t}|_{\mathbf{A}_t=\mathbf{I}}$ might be interpreted as Fisher information matrices evaluated at $\mathbf{A}_o = \mathbf{I} \in \mathcal{D}^N$ and $\mathbf{A}_t = \mathbf{I} \in \mathcal{D}^M$. Furthermore, if $\|\frac{\partial \mathcal{J}_{\ell_1}(\mathbf{Z})}{\partial \mathbf{A}_t}|_{\mathbf{A}_t=\mathbf{I}}\|_* = 0$ and $\|\frac{\partial \mathcal{J}_{\ell_1}(\mathbf{Y};\mathbf{Z})}{\partial \mathbf{A}_t}|_{\mathbf{A}_t=\mathbf{I}}\|_* = 0$, then $(R4)$ in information geometry is seen as change of coordinates on a manifold, where the intrinsic properties of curvature remain unchanged under different parametrization.

APPENDIX C.2

Note that for the model $\mathcal{P}^t$ we have that

$$
\begin{aligned}
\mathbf{y} &= \mathcal{T}(\mathbf{A}\mathbf{x}) = \max(\mathbf{A}\mathbf{x} - \boldsymbol{\tau}, \mathbf{0}) - \max(-\mathbf{A}\mathbf{x} - \boldsymbol{\tau}, \mathbf{0}), \\
\mathbf{q} &= \mathcal{T}(\mathbf{A}\mathbf{x}) = \max(\mathbf{A}\mathbf{g} - \boldsymbol{\tau}, \mathbf{0}) - \max(-\mathbf{A}\mathbf{g} - \boldsymbol{\tau}, \mathbf{0}),
\end{aligned} \tag{59}
$$

since

$$
\text{sign}(a)\max(|a| - b, 0) = \max(a - b, 0) - \max(-a - b, 0), \tag{60}
$$

The first order derivative of the divergence $D_{\ell_1}^{\mathcal{P}^t}(\mathbf{x}; \mathbf{g})$ w.r.t. the parameter $\mathbf{A}$ is:

$$
\begin{aligned}
\frac{\partial D_{\ell_1}^{\mathcal{P}^t}(\mathbf{x}; \mathbf{g})}{\partial \mathbf{A}} &= \\
\frac{\partial(\max(\mathbf{A}\mathbf{x} - \boldsymbol{\tau}, \mathbf{0})^T \max(\mathbf{A}\mathbf{g} - \boldsymbol{\tau}, \mathbf{0}))}{\partial \mathbf{A}} &+ \frac{\partial(\max(-\mathbf{A}\mathbf{x} - \boldsymbol{\tau}, \mathbf{0})^T \max(-\mathbf{A}\mathbf{g} - \boldsymbol{\tau}, \mathbf{0}))}{\partial \mathbf{A}}
\end{aligned} \tag{61}
$$

we assume that the threshold parameter $\boldsymbol{\tau}$ is chosen such that the vector $|\mathbf{A}\mathbf{x}| - \boldsymbol{\tau}$ (or for any other $\mathbf{q}$, the vector $|\mathbf{A}\mathbf{q}| - \boldsymbol{\tau}$ ) has least one non-zero element, then

$$
\frac{\partial(\max(\mathbf{A}\mathbf{x} - \boldsymbol{\tau}, \mathbf{0})^T \max(\mathbf{A}\mathbf{g} - \boldsymbol{\tau}, \mathbf{0}))}{\partial \mathbf{A}} = \max(\mathbf{A}\mathbf{g} - \boldsymbol{\tau}, \mathbf{0})\mathbf{x}^T + \max(\mathbf{A}\mathbf{x} - \boldsymbol{\tau}, \mathbf{0})\mathbf{g}^T, \tag{62}
$$

and

$$
\begin{aligned}
\frac{\partial(\max(-\mathbf{A}\mathbf{x} - \boldsymbol{\tau}, \mathbf{0})^T \max(-\mathbf{A}\mathbf{g} - \boldsymbol{\tau}, \mathbf{0}))}{\partial \mathbf{A}} &= \\
- \max(-\mathbf{A}\mathbf{g} - \boldsymbol{\tau}, \mathbf{0})\mathbf{x}^T &- \max(-\mathbf{A}\mathbf{x} - \boldsymbol{\tau}, \mathbf{0})\mathbf{g}^T,
\end{aligned} \tag{63}
$$

combining (62) and (63) we have that

$$
\begin{aligned}
\frac{\partial D_{\ell_1}^{\mathcal{P}^t}(\mathbf{x}; \mathbf{g})}{\partial \mathbf{A}} &= \\
\max(\mathbf{A}\mathbf{g} - \boldsymbol{\tau}, \mathbf{0})\mathbf{x}^T &+ \max(\mathbf{A}\mathbf{x} - \boldsymbol{\tau}, \mathbf{0})\mathbf{g}^T - \\
(\max(-\mathbf{A}\mathbf{g} + \boldsymbol{\tau}, \mathbf{0})\mathbf{x}^T &+ \max(-\mathbf{A}\mathbf{x} - \boldsymbol{\tau}, \mathbf{0})\mathbf{g}^T) = \\
\mathbf{q}\mathbf{x}^T &+ \mathbf{y}\mathbf{g}^T
\end{aligned} \tag{64}
$$

where

$$
\begin{aligned}
y(m) &= \text{sign}(\mathbf{a}_m^T \mathbf{x})\max(|\mathbf{a}_m^T \mathbf{x}| - \tau(m), 0) \\
q(m) &= \text{sign}(\mathbf{a}_m^T \mathbf{g})\max(|\mathbf{a}_m^T \mathbf{g}| - \tau(m), 0),
\end{aligned} \tag{65}
$$

$\forall m \in \mathcal{M}$.

Similarity, note that for the model $\mathcal{P}_0^o = \{\mathbf{A}_o, \boldsymbol{\tau} = \mathbf{0}\}$ we have that

$$
\begin{aligned}
\mathbf{x}_o =& \mathcal{T}^{\mathcal{P}_0^o}(\mathbf{A}_o \mathbf{x}) = \\
& \text{sign}(\mathbf{A}_o \mathbf{x}) \odot \max(|\mathbf{A}_o \mathbf{x}| - \mathbf{0}, \mathbf{0}) = \\
& \max(\mathbf{A}_o \mathbf{x} - \mathbf{0}, \mathbf{0}) - \max(-\mathbf{A}_o \mathbf{x} - \mathbf{0}, \mathbf{0}), \\
\mathbf{g}_o =& \mathcal{T}^{\mathcal{P}_0^o}(\mathbf{A}_o \mathbf{g}) = \\
& \text{sign}(\mathbf{A}_o \mathbf{g}) \odot \max(|\mathbf{A}_o \mathbf{g}| - \mathbf{g}, \mathbf{0}) = \\
& \max(\mathbf{A}_o \mathbf{g} - \mathbf{0}, \mathbf{0}) - \max(-\mathbf{A}_o \mathbf{g} - \mathbf{0}, \mathbf{0}).
\end{aligned}
\tag{66}
$$

The first order derivative of the divergence $D_{\ell_1}^{\mathcal{P}_0^o}(\mathbf{x}; \mathbf{g})$ w.r.t $\mathbf{A}_o$ is:

$$
\frac{\partial D_{\ell_1}^{\mathcal{P}_0^o}(\mathbf{x}; \mathbf{g})}{\partial \mathbf{A}_o} = \mathbf{x}_o \mathbf{g}^T + \mathbf{g}_o \mathbf{x}^T,
\tag{67}
$$

note that at $\mathbf{A}_o = \mathbf{I}$, $\mathcal{P}_0^t = \mathcal{B}^N$ and we have that

$$
\frac{\partial D_{\ell_1}^{\mathcal{B}^N}(\mathbf{x}; \mathbf{g})}{\partial \mathbf{A}_o}\Big|_{\mathbf{A}_o = \mathbf{I}} = \mathbf{x} \mathbf{g}^T + \mathbf{g} \mathbf{x}^T
\tag{68}
$$

Also for the model $\mathcal{P}_0^t = \{\mathbf{A}_t, \boldsymbol{\tau} = \mathbf{0}\}$ we have that

$$
\begin{aligned}
\mathbf{y}_t =& \mathcal{T}^{\mathcal{P}_0^t}(\mathbf{A}_t \mathbf{y}) = \\
& \text{sign}(\mathbf{A}_t \mathbf{y}) \odot \max(|\mathbf{A}_t \mathbf{y}| - \mathbf{0}, \mathbf{0}) = \\
& \max(\mathbf{A}_t \mathbf{y} - \mathbf{0}, \mathbf{0}) - \max(-\mathbf{A}_t \mathbf{y} - \mathbf{0}, \mathbf{0}), \\
\mathbf{q}_t =& \mathcal{T}^{\mathcal{P}_0^t}(\mathbf{A}_t \mathbf{q}) = \\
& \text{sign}(\mathbf{A}_t \mathbf{q}) \odot \max(|\mathbf{A}_t \mathbf{q}| - \mathbf{q}, \mathbf{0}) = \\
& \max(\mathbf{A}_t \mathbf{q} - \mathbf{0}, \mathbf{0}) - \max(-\mathbf{A}_t \mathbf{q} - \mathbf{0}, \mathbf{0}).
\end{aligned}
\tag{69}
$$

The first order derivative of the divergence $D_{\ell_1}^{\mathcal{P}_0^t}(\mathbf{y}; \mathbf{q})$ w.r.t $\mathbf{A}_t$ is:

$$
\frac{\partial D_{\ell_1}^{\mathcal{P}_0^t}(\mathbf{y}; \mathbf{q})}{\partial \mathbf{A}_t} = \mathbf{y}_t \mathbf{q}^T + \mathbf{q}_t \mathbf{y}^T,
\tag{70}
$$

note that at $\mathbf{A}_t = \mathbf{I}$, $\mathcal{P}_0^t = \mathcal{B}^M$ and we have that

$$
\frac{\partial D_{\ell_1}^{\mathcal{B}^M}(\mathbf{y}; \mathbf{q})}{\partial \mathbf{A}_t}\Big|_{\mathbf{A}_o = \mathbf{I}} = \mathbf{y} \mathbf{q}^T + \mathbf{q} \mathbf{y}^T
\tag{71}
$$

Consider the following

$$
\begin{aligned}
\mathbf{A}\mathbf{g} &= \mathbf{q} + \mathbf{z}_1 / \mathbf{x}^T \\
\mathbf{A}\mathbf{x} &= \mathbf{y} + \mathbf{z}_2 / \mathbf{g}^T
\end{aligned}
\rightarrow
\begin{cases}
\mathbf{A}\mathbf{x}\mathbf{g}^T = \mathbf{y}\mathbf{g}^T + \mathbf{z}_1 \mathbf{g}^T \\
\mathbf{A}\mathbf{g}\mathbf{x}^T = \mathbf{q}\mathbf{x}^T + \mathbf{z}_2 \mathbf{x}^T
\end{cases}
$$
$$
\mathbf{A}(\mathbf{x}\mathbf{g}^T + \mathbf{g}\mathbf{x}^T) = \mathbf{y}\mathbf{g}^T + \mathbf{q}\mathbf{x}^T + \mathbf{z}_1 \mathbf{g}^T + \mathbf{z}_2 \mathbf{x}^T
\tag{72}
$$

where

$$
\begin{aligned}
\mathbf{z}_1 &= \mathbf{A}\mathbf{x} - \text{sign}(\mathbf{A}\mathbf{x}) \max(|\mathbf{A}\mathbf{x}| - \tau\mathbf{1}, \mathbf{0}) \\
\mathbf{z}_2 &= \mathbf{A}\mathbf{g} - \text{sign}(\mathbf{A}\mathbf{g}) \max(|\mathbf{A}\mathbf{g}| - \tau\mathbf{1}, \mathbf{0})
\end{aligned}
\tag{73}
$$

A closer look at (72) reveals us that

$$
\mathbf{A}\frac{\partial D_{\ell_1}^{\mathcal{B}^N}(\mathbf{x}; \mathbf{g})}{\partial \mathbf{A}_o}\Big|_{\mathbf{A}_o = \mathbf{I}} = \frac{\partial D_{\ell_1}^{\mathcal{P}}(\mathbf{x}; \mathbf{g})}{\partial \mathbf{A}} + \delta_o^{z_1, z_2}
\tag{74}
$$

where

$$\boldsymbol{\delta}_o^{z_1, z_2} = \mathbf{z}_1 \mathbf{g}^T + \mathbf{z}_2 \mathbf{x}^T \tag{75}$$

By similar construction applied to the rest of the pairs of data samples we have the result:

$$\mathbf{A} \left( \frac{\partial D_{\ell_1, c}^{\mathcal{B}^N}(\mathbf{X})}{\partial \mathbf{A}_o} |_{\mathbf{A}_o = \mathbf{I}} + \frac{\partial D_{\ell_1}^{\mathcal{B}^N}(\mathbf{X})}{\partial \mathbf{A}_o} |_{\mathbf{A}_o = \mathbf{I}} \right) = \frac{\partial D_{\ell_1, c}^{\mathcal{P}}(\mathbf{X})}{\partial \mathbf{A}} + \frac{\partial D_{\ell_1}^{\mathcal{P}}(\mathbf{X})}{\partial \mathbf{A}} + \boldsymbol{\delta}_o \square \tag{76}$$

Note that

$$\begin{aligned} \mathbf{A}\mathbf{x} &= \mathbf{y} + \mathbf{z}_1/\mathbf{q}^T \\ \mathbf{A}\mathbf{g} &= \mathbf{q} + \mathbf{z}_2/\mathbf{y}^T \end{aligned} \rightarrow \left\{ \begin{aligned} \mathbf{A}\mathbf{x}\mathbf{q}^T &= \mathbf{y}\mathbf{q}^T + \mathbf{z}_1\mathbf{q}^T \\ \mathbf{A}\mathbf{g}\mathbf{y}^T &= \mathbf{q}\mathbf{y}^T + \mathbf{z}_2\mathbf{y}^T \end{aligned} \right. \tag{77}$$
$$\mathbf{A}(\mathbf{x}\mathbf{y}^T + \mathbf{q}\mathbf{g}^T) = \mathbf{y}\mathbf{q}^T + \mathbf{q}\mathbf{y}^T + \mathbf{z}_1\mathbf{q}^T + \mathbf{z}_2\mathbf{y}^T,$$

where

$$\begin{aligned} \mathbf{z}_1 &= \mathbf{A}\mathbf{x} - \operatorname{sign}(\mathbf{A}\mathbf{x}) \max(|\mathbf{A}\mathbf{x}| - \tau\mathbf{1}, \mathbf{0}) \\ \mathbf{z}_2 &= \mathbf{A}\mathbf{g} - \operatorname{sign}(\mathbf{A}\mathbf{g}) \max(|\mathbf{A}\mathbf{g}| - \tau\mathbf{1}, \mathbf{0}) \end{aligned} \tag{78}$$

A closer look at (77) reveals us

$$\mathbf{A} \left( \frac{\partial D_{\ell_1}^{\mathcal{P}}(\mathbf{x}; \mathbf{g})}{\partial \mathbf{A}} \right)^T = \frac{\partial D_{\ell_1}^{\mathcal{B}^M}(\mathbf{y}; \mathbf{q})}{\partial \mathbf{A}_t} |_{\mathbf{A}_t = \mathbf{I}} + \boldsymbol{\delta}_t^{z_1, z_2}. \tag{79}$$

where

$$\boldsymbol{\delta}_t^{z_1, z_2} = \mathbf{z}_1 \mathbf{q}^T + \mathbf{z}_2 \mathbf{y}^T \tag{80}$$

By similar construction applied to the rest of the pairs of data samples we have the result:

$$\mathbf{A} \left( \frac{\partial D_{\ell_1, c}^{\mathcal{P}^t}(\mathbf{X})}{\partial \mathbf{A}} + \frac{\partial D_{\ell_1}^{\mathcal{P}^t}(\mathbf{X})}{\partial \mathbf{A}} \right)^T = \left( \frac{\partial D_{\ell_1, c}^{\mathcal{B}^M}(\mathbf{Y})}{\partial \mathbf{A}_t} |_{\mathbf{A}_t = \mathbf{I}} + \frac{\partial D_{\ell_1}^{\mathcal{B}^M}(\mathbf{Y})}{\partial \mathbf{A}_t} |_{\mathbf{A}_t = \mathbf{I}} \right) + \boldsymbol{\delta}_t \square \tag{81}$$

Note that

$$\begin{aligned} \mathbf{A}\mathbf{x} &= \mathbf{y} + \mathbf{z}_1/\mathbf{z}_2^T \\ \mathbf{A}\mathbf{g} &= \mathbf{q} + \mathbf{z}_2/\mathbf{z}_1^T \end{aligned} \rightarrow \left\{ \begin{aligned} \mathbf{A}\mathbf{x}\mathbf{z}_2^T &= \mathbf{y}\mathbf{z}_2^T + \mathbf{z}_1\mathbf{z}_2^T \\ \mathbf{A}\mathbf{g}\mathbf{z}_1^T &= \mathbf{q}\mathbf{z}_1^T + \mathbf{z}_2\mathbf{z}_1^T \end{aligned} \right. \tag{82}$$
$$\mathbf{A}(\mathbf{x}\mathbf{z}_2^T + \mathbf{q}\mathbf{z}_1^T) = \mathbf{y}\mathbf{z}_2^T + \mathbf{q}\mathbf{z}_1^T + \mathbf{z}_1\mathbf{z}_2^T + \mathbf{z}_2\mathbf{z}_1^T,$$

where

$$\begin{aligned} \mathbf{z}_1 &= \mathbf{A}\mathbf{x} - \operatorname{sign}(\mathbf{A}\mathbf{x}) \max(|\mathbf{A}\mathbf{x}| - \tau\mathbf{1}, \mathbf{0}) \\ \mathbf{z}_2 &= \mathbf{A}\mathbf{g} - \operatorname{sign}(\mathbf{A}\mathbf{g}) \max(|\mathbf{A}\mathbf{g}| - \tau\mathbf{1}, \mathbf{0}) \end{aligned} \tag{83}$$

A closer look at (82) reveals us

$$\mathbf{A}(\boldsymbol{\delta}_o^{z_1, z_2})^T = \frac{\partial D_{\ell_1}^{\mathcal{B}^M}(\mathbf{z}_1; \mathbf{z}_2)}{\partial \mathbf{A}_t} |_{\mathbf{A}_t = \mathbf{I}} + (\boldsymbol{\delta}_t^{z_1, z_2})^T. \tag{84}$$

where

$$\boldsymbol{\delta}_t^{z_1, z_2} = \mathbf{z}_1 \mathbf{q}^T + \mathbf{z}_2 \mathbf{y}^T \tag{85}$$

By similar construction applied to the rest of the pairs of data samples we have the result:

$$\mathbf{A}\boldsymbol{\delta}_o^T = \left( \frac{\partial D_{\ell_1, c}^{\mathcal{B}^M}(\mathbf{Z})}{\partial \mathbf{A}_t} |_{\mathbf{A}_t = \mathbf{I}} + \frac{\partial D_{\ell_1}^{\mathcal{B}^M}(\mathbf{Z})}{\partial \mathbf{A}_t} |_{\mathbf{A}_t = \mathbf{I}} \right) + \boldsymbol{\delta}_t^T \square \tag{86}$$

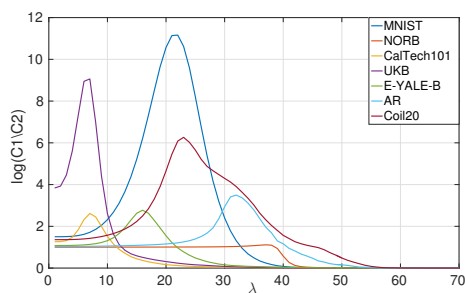 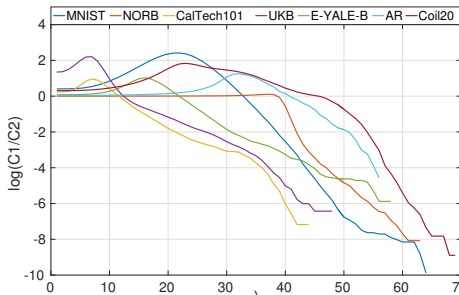

Figure 9: The ratio $C1/C2$ of the similarity concentrations $C1 = D^{\mathcal{B}^M}_{\ell_1,c}(\mathbf{Y})$ and $C2 = D^{\mathcal{B}^M}_{\ell_1}(\mathbf{Y})$ and the discrimination power $\log(C1/C2) = \mathcal{I}^t$ for randomly chosen subsets from all of the used databases under a non-linear transform with transform dimension $M = 19000$ and varying thresholding parameter $\boldsymbol{\tau} = \lambda\mathbf{1}$.

## Appendix D

The result in by *Lemma* **2** (58) decomposes on the contribution components for simmilarity and on the contributiong components for dissimilarity

$$\frac{\partial \mathcal{J}_{\ell_1}(\mathbf{AX})}{\partial \mathbf{A}_t}|_{\mathbf{A}_t=\mathbf{I}} = \frac{\partial \mathcal{J}_{\ell_1}(\mathbf{AX})}{\partial \mathbf{A}_t}|_{\mathbf{A}_t=\mathbf{I}}|_s - \frac{\partial \mathcal{J}_{\ell_1}(\mathbf{AX})}{\partial \mathbf{A}_t}|_{\mathbf{A}_t=\mathbf{I}}|_d + \boldsymbol{\xi}_c|_s - \boldsymbol{\xi}_c|_d + \boldsymbol{\xi}|_s - \boldsymbol{\xi}|_d. \tag{87}$$

Moreover, w.r.t. the simmilarity concentrations we have the following decompositions

$$Tr\{\frac{\partial D^{\mathcal{B}^M}_{\ell_1,c}(\mathbf{AX})}{\partial \mathbf{A}_t}|_{\mathbf{A}_t=\mathbf{I}}\} = D^{\mathcal{B}^M}_{\ell_1,c}(\mathbf{Y}) + Tr\{\boldsymbol{\xi}_c|_s\} = D^{\mathcal{P}}_{\ell_1,c}(\mathbf{X}) + Tr\{\boldsymbol{\xi}_c|_s\}$$

$$Tr\{\frac{\partial D^{\mathcal{B}^M}_{\ell_1}(\mathbf{AX})}{\partial \mathbf{A}_t}|_{\mathbf{A}_t=\mathbf{I}}\} = D^{\mathcal{B}^M}_{\ell_1}(\mathbf{Y}) + Tr\{\boldsymbol{\xi}|_s\} = D^{\mathcal{P}}_{\ell_1}(\mathbf{X}) + Tr\{\boldsymbol{\xi}|_s\}, \tag{88}$$

therfore we have the following bounds

$$a : Tr\{\mathbf{A}\frac{\partial D^{\mathcal{B}^N}_{\ell_1,c}(\mathbf{X})}{\partial \mathbf{A}_o}|_{\mathbf{A}_o=\mathbf{I}}\mathbf{A}^T\} \le D^{\mathcal{P}}_{\ell_1,c}(\mathbf{X}) \le Tr\{\frac{\partial D^{\mathcal{B}^M}_{\ell_1,c}(\mathbf{AX})}{\partial \mathbf{A}_t}|_{\mathbf{A}_t=\mathbf{I}}|_s\} = D^{\mathcal{B}^M}_{\ell_1,c}(\mathbf{AX})$$

$$b : Tr\{\mathbf{A}\frac{\partial D^{\mathcal{B}^N}_{\ell_1}(\mathbf{X})}{\partial \mathbf{A}_o}|_{\mathbf{A}_o=\mathbf{I}}\mathbf{A}^T\} \le D^{\mathcal{P}}_{\ell_1}(\mathbf{X}) \le Tr\{\frac{\partial D^{\mathcal{B}^M}_{\ell_1}(\mathbf{AX})}{\partial \mathbf{A}_t}|_{\mathbf{A}_t=\mathbf{I}}|_s\} = D^{\mathcal{B}^M}_{\ell_1}(\mathbf{AX}) \tag{89}$$

Additionaly, we have that $c : \lambda_{min}(\mathbf{A}^T\mathbf{A})Tr\{\frac{\partial D^{\mathcal{B}^N}_{\ell_1,c}(\mathbf{X})}{\partial \mathbf{A}_o}|_{\mathbf{A}_o=\mathbf{I}}\} \le Tr\{\mathbf{A}\frac{\partial D^{\mathcal{B}^N}_{\ell_1,c}(\mathbf{X})}{\partial \mathbf{A}_o}|_{\mathbf{A}_o=\mathbf{I}}\mathbf{A}^T\}$, where $\lambda_{min}(\mathbf{A}^T\mathbf{A})$ is the minimum singlular value of the matrix $\mathbf{A}^T\mathbf{A}$. Taking the logarithm of the ratio $\frac{D^{\mathcal{P}}_{\ell_1,c}(\mathbf{X})}{D^{\mathcal{P}}_{\ell_1}(\mathbf{X})+\epsilon}$ using the bounds $a$, $b$ and $c$ we arrive at the desired result $\square$

## Appendix E

The exact steps of the proposed non-linear transform learning are described by Algorithm 1.

## Appendix F

The ratio $C1/C2$ between the similarity concentrations $C1 = D^{\mathcal{B}^M}_{\ell_1,c}(\mathbf{Y})$ and $C2 = D^{\mathcal{B}^M}_{\ell_1}(\mathbf{Y})$ and the discrimination power $\log(C1/C2) = \mathcal{I}^t$ on subsets of the used databases after applying a non-linear transform with transform dimension $M = 19000$ and varying the thresholding parameter $\boldsymbol{\tau} = \lambda\mathbf{1}$ is shown in Figure 9. We used 70 different values for the parameter $\lambda$, sampled uniformly from the interval $\left(0, (\max_{1 \le c \le C, 1 \le k \le K} \max_{1 \le m \le M} |\mathbf{a}_m^T\mathbf{x}_{c,k}|)\right)$. The results were obtained using

---

**Algorithm 1** Non-linear transform learning algorithm

---

**Input** $\mathbf{X}, \lambda_0, \lambda_1, \lambda_2, \lambda_3, \lambda_4$
$\mathbf{A} \leftarrow inicialize$
**repeat**
    DISCRIMINATIVE ENCODING closed form solution per data sample
        $\mathbf{Y} \leftarrow \mathbf{AX}$
        **repeat**
            **for** $\forall c \in \mathcal{C}$ **do**
                $\mathbf{d}_c^- \leftarrow \sum_{\substack{c1 \\ c1 \neq c}} \sum_{k1} \mathbf{y}_{c1,k1}^-, \mathbf{d}_c^+ \leftarrow \sum_{\substack{c1 \\ c1 \neq c}} \sum_{k1} \mathbf{y}_{c1,k1}^+$ and
                $\mathbf{s}_c \leftarrow \sum_{\substack{c1 \\ c1 \neq c}} \sum_{k1} \mathbf{y}_{c1,k1} \odot \mathbf{y}_{c1,k1}$
            **end for**
            **for** $\forall c \in \mathcal{C}$ and $\forall k \in \mathcal{K}$ **do**
                $\mathbf{g} \leftarrow \text{sign}(\max(\mathbf{Ax}_{c,k}, \mathbf{0})) \odot \mathbf{d}_c^+ + \text{sign}(\max(-\mathbf{Ax}_{c,k}, \mathbf{0})) \odot \mathbf{d}_c^-$
                $\mathbf{y}_{c,k} \leftarrow \text{sign}(\mathbf{Ax}_{c,k}) \odot \max(\mathbf{Ax}_{c,k} - \lambda_0 \mathbf{g} + \lambda_1 \mathbf{1}, \mathbf{0}) \oslash (1 + 2\lambda_0 \mathbf{s}_c)$
            **end for**
        **until** $convergence$
    TRANSFORM UPDATE $\epsilon$-close closed form solution
        $\mathbf{U}_X \mathbf{\Sigma}_X^2 \mathbf{U}_X^T \leftarrow \mathbf{XX}^T + \lambda_2 \mathbf{I}$ and $\mathbf{U}_{U_X XY} \mathbf{\Sigma}_{U_X XY} \mathbf{V}_{U_X XY}^T \leftarrow \mathbf{U}_X^T \mathbf{XY}^T$
        $\min_{\sigma_A(n)} \frac{\lambda_3}{\sigma_X^4} \sigma_A^4(n) + \left( \frac{\sigma_X^2(n) - 2\lambda_3}{\sigma_X^2(n)} \right) \sigma_A^2(n) - \frac{\sigma_\Gamma(n)}{\sigma_X(n)} \sigma_A(n) - 2\lambda_4 \log \frac{\sigma_A(n)}{\sigma_X(n)}$
            where $\sigma_\Gamma(n) \leftarrow T(n,n), \mathbf{T} \leftarrow \mathbf{U}_{U_X XY} \mathbf{\Sigma}_{U_X XY} \mathbf{U}_{U_X XY}^T, \forall n \in \mathcal{N}$
        $\mathbf{A} \leftarrow \mathbf{V}_{U_X XY} \mathbf{U}_{U_X XY}^T \mathbf{\Sigma}_A \mathbf{\Sigma}_X^{-1} \mathbf{U}_X^T$
**until** $convergence$
**Output** $\mathbf{A}, \mathbf{Y}$

---

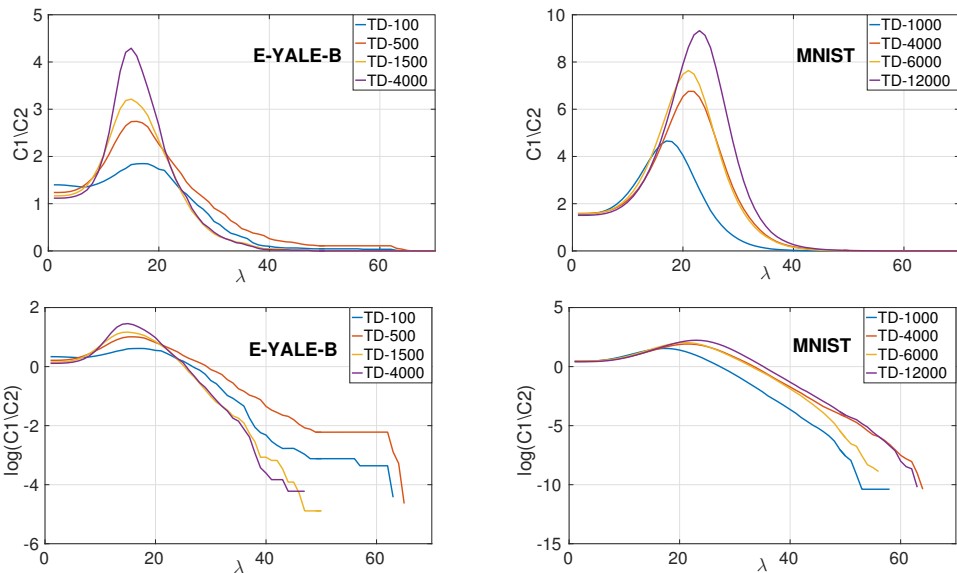

Figure 10: The ratio of the similarity concentrations similarity concentrations $C1 = D_{\ell_1,c}^{\mathcal{B}^M}(\mathbf{Y})$ and $C2 = D_{\ell_1}^{\mathcal{B}^M}(\mathbf{Y})$ and the discrimination power $\log(C1/C2) = \mathcal{I}^t$ for the Extended Yale B and MNIST databases under non-linear transforms having different transform dimension $M$ and varying thresholding parameter $\boldsymbol{\tau} = \lambda \mathbf{1}$.

a non-linear transform learned with one value for parameter $\lambda$ for all the databases. Since all the databases have different variabilities and the amount of available data is different, this result suggest that per different database there should be different optimal values for the parameter $\lambda$.

The results about the ratio $C1/C2$ between the similarity concentrations $C1 = D_{\ell_1,c}^{\mathcal{B}^M}(\mathbf{Y})$ and $C2 = D_{\ell_1}^{\mathcal{B}^M}(\mathbf{Y})$ and the discrimination power $\log(C1/C2) = \mathcal{I}^t$ on the Extanded-Yale-B and the MNIST data sets after applying a non-linear transform with transform dimensions $M = \{100, 500, 1500, 4000\}$ and $M = \{1000, 4000, 6000, 12000\}$, and varying the thresholding parameter $\boldsymbol{\tau} = \lambda \mathbf{1}$ are shown in Figure 10. We again used 70 different values for the parameter $\lambda$, sampled uniformly from the interval $\left(0, (\max_{1 \leq c \leq C, 1 \leq k \leq K} \max_{1 \leq m \leq M} |\mathbf{a}_m^T \mathbf{x}_{c,k}|)\right)$. The results were obtained using a non-linear transforms learned with optimally choose values (by using cross-validation) of the parameter $\lambda$ for the two different databases. As expected, we can see that the extreme points of the ratio between the $\ell_*$-norms $C1$ and $C2$ of the similarity concentrations and the discrimination power is around the optimal values of the parameter $\boldsymbol{\tau} = \lambda \mathbf{1}$.

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
