# OpenReview forum: "Learning non-linear transform with discriminative and minimum information loss priors"
_ICLR.cc/2018/Conference — Reject_

### Official Review · AnonReviewer1 · 2017-11-26
**A potentially interesting idea, but the paper is a bit hard to follow**

**Rating:** 4
**Confidence:** 2

**Review:**

Summary:
The paper proposes a model to estimate a non-linear transform of data with labels, trying to increase the discriminative power of the transformation while preserving information.

Quality:
The quality is potentially good but I misunderstood too many things (see below) to emit a confident judgement.

Clarity:
Clarity is poor. The paper is overall difficult to follow (at least for me) for different reasons. First, with 17 pages + references + appendix, the length of the paper is way above the « strongly suggested limit of 8 pages ». Second, there are a number of small typos and the citations are not well formatted (use \citep instead of \citet). Third, and more importantly, several concepts are only vaguely formulated (or wrong), and I must say I certainly misunderstood some parts of the manuscript.
A few examples of things that should be clarified:
- p4: « per class c there exists an unknown nonlinear function […] that separates the data samples from different classes in the transformed domain ». What does « separate » mean here? There exists as many functions as there are classes, do you mean that each function separates all classes? Or that when you apply each class function to the elements of its own class, you get a separation between the classes? (which would not be very useful)
- p5: what do you mean exactly by « non-linear thresholding function »?
- p5: « The goal of learning a nonlinear transform (2) is to estimate… »: not sure what you mean by « accurate approximation » in this sentence.
- p5 equation 3: if I understand correctly, you not only want to estimate a single vector of thresholds for all classes, but also want it to be constant. Is there a reason for the second constraint?
- p5 After equation 4, you define different « priors ». The one on z_k seems to be Gaussian; however in equation 4, z_k seems to be the difference between the input and the output of the nonlinear threshold operator. If this is correct, then the nonlinear threshold operator is just the addition of a Gaussian random noise, which is really not a thresholding operator. I suppose I misunderstood something here, some clarification is probably needed.
- p5 before equation 5, you define a conditional distribution of \tau_c given y_{c,k} ; however if you define a prior on \tau_c given each point (i.e., each $k$), how do you define the law of \tau_c given all points?
- p5 Equation 6: I dont understand what the word « approximation » refers to, and how equation 6 is derived.
-p6 equation 7 : missing exponent in the Gaussian distribution
- p6-7: to combine equations 7-8-9 and obtain 10, I suppose in equation 7 the distributions should be conditioned on A (at least the first one), and in equation 9 I suppose the second line should be removed and the third line should be conditioned on A; otherwise more explanations are needed.
- Lemma 1 is just unreadable. Please at least split equations in several lines.

Originality:
As far as I can tell, the approach is quite original and the results proved are new.

Significance:
The method provides a new way to learn discriminative features; as such it is a variant of several existing methods, which could have some impact if clarity is improved and a public code is provided.

---

> ### Author Response · Authors · 2017-12-04
> **Comment to the reviewer**
>
> We would like to thank the reviewer for the time spent on the detailed and careful reading of our paper and providing his valuable comments.
>
> We agree with the reviewer that the presentation quality should be improved and we will do our best to improve it. The concepts in section 1.2 paragraph one, the introduction of section 2 and section 2.1 will be simplified and better clarified.
>
> Considering the typos, reformat of the citations and the clarification we will try definitively to enhance them accordingly.
>
> Considering the length of the paper, we can integrate the comment of the reviewer and add the sensitivity analysis section in the appendix. However, since the results of the analysis are used to bound the proposed discrimination power measure we considered not to suspend and remove them. In addition they also give an information-geometric perspective, that to the best of our knowledge is the first analysis of this kind about a non-linear models and the similarity concentration measure without the need of a strict conditions for regularity, i.e., smoothness of the manifolds.
>
>
> Considering the comment "p4" and the first 3 "p5" comments
>
> We will clarify here and in the updated version that a non-linear transform models having one common ${\bf A}$ and different thresholding parameter ${\boldsymbol{\tau}_c}$ per class $c$ are used only during the learning, whereas at the test time only a sparsifying transform model is used. In that sense we assumed that there are number of non-linear transforms as the number of classes and when we apply the non-linear transforms for the samples of the corresponding classes we have separation between the transform data samples from different classes. It is important to highlight that when the proposed similarity concentration is zero then the discriminative prior has no influence. Only then the non-linear transform model reduces to a sparsifying transform model and the single vector of thresholds is a constant.
>
> Considering the comment "p5 After equation 4 ..."
>
> We use a transform representation defined as ${\bf y}_{c,k}=\mathcal{T}^{\mathcal{P}_c}({\bf x}_{c,k})$ to impose a dicriminative constraint on the transform representation. As shown by equation (16) the estimation of the transform representation has a closed form solution considering a thresholding non-linear operation, therefore, it has
> a non-linearity. Moreover, in this case ${\bf A}{\bf x}_{c,k}$ is only seen as a linear approximation to this non-linearity. Knowing something in advance about the difference between ${\bf y}_{c,k}-{\bf A}{\bf x}_{c,k}$ can be used in our model. However, since in advance we do not have any prior we assume that it is Gausssian like distributed.
>
>
> Considering the comment "p5 before equation 5, you define ..."
>
> We assumed that we have a joint probability $p(\boldsymbol{\tau}_1, \boldsymbol{\tau}_2,...,\boldsymbol{\tau}_C, {\bf y}_{c,k})=p(\boldsymbol{\tau}_1, \boldsymbol{\tau}_2,...,\boldsymbol{\tau}_C|{\bf y}_{c,k})p({\bf y}_{c,k}) \propto exp(-\frac{ min_{1 \leq c \leq C}( D({\boldsymbol{\tau}_c; {\bf y}_{c,k}}) )   }{  \beta_2 })exp(-\frac{ \Vert {\bf y}_{c,k}  \Vert_1   }{  \beta_0 })$. If we
>
> further assume that $p(\boldsymbol{\tau}_1, \boldsymbol{\tau}_2, ..., \boldsymbol{\tau}_C)=\prod_{c=1}^C p(\boldsymbol{\tau}_c)$ and that the class label is know then we say that
>
> $p(\boldsymbol{\tau}_c| {\bf y}_{c,k})  \propto exp(-\frac{D({\boldsymbol{\tau}_c; {\bf y}_{c,k}} )}{\beta_2})$.
>
>
> Considering the comment "p5 Equation 6: I dont ..." we note that there was a typo (a summation is missing). That is
> $D({\bf y}_{c,k}; \boldsymbol{\tau}_{c} ) $ is not equal to $D^\mathcal{P}_{\ell_1} ({\bf X}) + S^\mathcal{P}_{\ell_2} ({\bf X})$. Since a summation in front of $D({\bf y}_{c,k}; \boldsymbol{\tau}_{c} ) $ in equation (6) was missing the correct expression for the first line in equation (6) is:
>
> $\sum_c D({\bf y}_{c,k}; \boldsymbol{\tau}_{c} ) =D^\mathcal{P}_{\ell_1} ({\bf X}) + S^\mathcal{P}_{\ell_2} ({\bf X})$
>
>
> Considering the comments p6-7 and Lemma 1 and all the former about the clarification pointed by the reviewer will be taken into considerations, we are working on the improvements that will be added into the revised version.
>
> We share the reviewers concern about the necessity to provide a public code to boost the impact of the work. However, while we emphasize that the provided pseudo-code in the paper is very clear and easily reproducible, our code will shortly be accessible to public short after the announcement of the official decision. We simply do not publish it now in order to maintain the anonymity of this review procedure. Moreover, addressing the reviewers concern about the significance and impact, we highly appreciate any concrete suggestion to improve, remaining open for any suggestions from the reviewers.

---

### Official Review · AnonReviewer3 · 2017-11-27
**Long paper with good proofs**

**Rating:** 5
**Confidence:** 2

**Review:**

This paper proposes a method of learning sparse dictionary learning by introducing new types of priors. Specifically, they designed a novel idea of defining a metric to measure discriminative properties along with the quality of presentations.
It is also presented the power of the proposed method in comparison with the existing methods in the literature.

Overall, the paper deals with an important issue in dictionary learning and proposes a novel idea of utilizing a set of priors.

To this reviewer’s understanding, the thresholding parameter $\tau_{c}$ is specific for a class $c$ only, thus different classes have different $\tau$ vectors. If so, Eq. (6) for approximation of the measure $D(\cdot)$ is not clear how the similarity measure between ${\bf y}_{c,k}$ and ${\bf y}_{c1,k1}$, \ie, $\left\|{\bf y}_{c,k}^{+}\odot{\bf y}_{c1,k1}^{+}\right\|_{1}+\left\|{\bf y}_{c,k}^{+}\odot{\bf y}_{c1,k1}^{+}\right\|_{1}$ and $\left\|{\bf y}_{c,k}\odot{\bf y}_{c1,k1}\right\|_{2}^{2}$, works to approximate it. It would be appreciated to give more detailed description on it and geometric illustration, if possible.

There are many typos and grammatical errors, which distract from reading and understanding the manuscript.

---

> ### Author Response · Authors · 2017-12-04
> **Comment to the reviewer**
>
> We would like to thank the reviewer for the time spend on the detailed, careful reading of our paper, providing his comments and the positive evaluation.
>
> Considering the comment about how the similarity measure works to approximate the measure in the prior, we note that there was a typo (a summation is missing). That is $D({\bf y}_{c,k}; \boldsymbol{\tau}_{c} ) $ is not equal to $D^\mathcal{P}_{\ell_1} ({\bf X}) + S^\mathcal{P}_{\ell_2} ({\bf X})$. Since a summation in front of $D({\bf y}_{c,k}; \boldsymbol{\tau}_{c} ) $ in equation (6) was missing the correct expression for the first line in equation (6) is:
>
> $\sum_c D({\bf y}_{c,k}; \boldsymbol{\tau}_{c} ) =D^\mathcal{P}_{\ell_1} ({\bf X}) + S^\mathcal{P}_{\ell_2} ({\bf X})  $
>
> Considering the geometrical illustration about the prior we had one, but, due to the long length of the paper we decided not to include it. However, since it was mentioned we will also take this into considerations.
>
> We will consider all the typos and grammatical errors and we will do our best to increase the presentation quality.

---

> ### Author Response · Authors · 2018-01-13
> **We would kindly like to ask the reviewer if he could comment on the reasons behind changing his reviews score.**
>
> We would like to extend the appreciation for taking the necessary time, involvement and effort in reading our initial and rebutted paper version, together with all the considerations, raised comments and concerns about all the aspects of this paper.
>
> If possible we would kindly like to ask the reviewer if he could comment on the reasons related to the current manuscript version that lead to changing his reviews score.

---

### Official Review · AnonReviewer2 · 2017-11-27
**A paper on representation learning with non-linear transform. Presentation quality should be improved.**

**Rating:** 5
**Confidence:** 1

**Review:**

Overview:
This paper proposes a method for learning representations using a “non-linear transform”. Specifically, the approach is based on the form: Y =~ AX, where X is the original data, A is a projection matrix, and Y is the resulting representation. Using some assumptions, and priors/regularizers on Y and A, a joint objective is derived (eq. 10), and an alternating optimization algorithm is proposed (eq. 11 and 14). Both objective and algorithm use approximations due to hardness of the problem. Theoretical and empirical results on the quality and properties of the representation are presented.
Disclaimer: this is somewhat outside my area of expertise, so this is a rather high-level review. I have not thoroughly checked proofs and claims.

Comments:
-I found the presentation quality to be rather poor, making it hard to fully understand and evaluate the approach. In particular, the motivation and approach are not clear (sec. 1.2), making it hard to understand the proposed method. There is no explicit formulation, instead there are references to other models (e.g., sparsifying transform model) and illustrative figures (fig. 1 and 2). Those are useful following a formal definition, but cannot replace it. The separation between positive and negative elements of the representation is not motivated and explained in a footnote although it seems central to the proposed approach.
- The paper is 17 pages long (24 pages with the appendix), so I had to skim through some parts. Due to the extensive scope, perhaps a journal submission would be more appropriate.

Minors:
- Vu & Monga 2016b and 2016c are the same.
- p. 1: meaner => manner
- p. 1: refereed => referred
- p. 1: “a structural constraints”; p. 2: “a low rank constraints”, “a pairwise constraints”; p. 4: “a similarity concentrations”, “a numerical experiments”, and others...
- p. 2, 7: therms => terms
- p. 2: y_{c_1,k_2} => y_{c_1,k_1}?
- p. 3, 4: “a the”
- p. 5: “an parametric”
- p. 8: ether => either
- Other typos… the paper needs proofreading.

---

> ### Author Response · Authors · 2017-12-04
> **The central motivation is given in section 1.2**
>
> We would like to thank the reviewer for the time spend on reading our paper and providing his comments.
>
>
> To the best of our knowledge this is the first attempt at extending the sparsifying transform model as a non-linear transform model. Moreover, we will clarify, simplify and highlight here, and in the revised version,  that a non-linear transform model is addressed only during the learning, whereas at test time only a sparsifying transform model is used. This is  explained by the fact that if the proposed similarity concentration  (that approximates the used measure in the discriminative
> prior) is zero then the discriminative prior is non-informative. Meaning  that the non-linear transform model reduces to a sparsifying transform model and the prior has no influence on the estimation of the representation. This  can easily be seen from the closed form solution of the transform representation in equation 16, that is if $ {\bf g}={\bf 0}$, then the transform representation is only a sparse representation, i.e., a thresholded version of ${\bf A}{\bf x}_{c,k}$.
>
> Yes, we agree that the motivation behind the approach is very important, therefore, in the current version we had
>
> section 1.2
>
> By the end of the first sentence in section 1.2 we meant that we do not address an inverse problem where if the dimensionality of the dictionary (transform matrix) or the data  is high then the solution of the inverse problem has a high computational complexity. Considering the estimation of the transform representation we radder address a direct problem that as pointed out in section 2.2 (before equation 15) represents a low complexity constrained projection problem and has a closed form solution (equation 16). By the end of the second sentence in section  1.2 in we reefer to the fact that the non-linear transform model allows more freedom in modeling and imposing constraints on the transform representation (in fact it allows other non-linearity to be modeled, i.e. very easily we can model a ReLu as a transform representation).
>
>
> Considering the motivation about the proposed central prior and the used approximation on the measure defined in the prior, first we note that we have tried to the best of our knowledge in sec 1.1 to outlay the open issues and the disadvantages in the specifics of the used constraints in the state-of-the-art discrimnative dictionary learning methods. Upon that we have devoted the second paragraph of section 1.2 on the general advantages and the advantages w.r.t. the state-of-the-art. More precisely we had:
>
>
> section 1.2 paragraph 2
>
>
> Additionally, the advantages of using this prior w.r.t. the discriminative priors in the state-of-the-art methods was given in
>
>
> section 2.1 last paragraph.
>
> We will try to clarify, restructure and highlight these points in our revised version.
>
> Concerning all the rest comments and the typos we will correct them accordingly.
>
> Considering the length of the paper, we note that there is a possibility the paper to be without the sensitivity analysis, moreover we can integrate the comment of the reviewer and add this section in the appendix. Nevertheless, we considered that the sensitivity analysis is beneficial since it is related to the notion about the quality of the representation, i.e., the results of the analysis are used to bound the proposed discrimination power measure. In addition they also give an information-geometric perspective, that also to the best of our knowledge is the first analysis of this kind about a non-linear models and the similarity concentration measure without the need of a strict conditions for regularity, i.e., smoothness of the manifolds.
>
> Since we target a representation learning we considered that ICLR is the best place to present  our work. We will try to reduce the paper length within the allowed limits of change w.r.t. the initial version and do our best to sharpen the quality of the presentation.

---

### Author Response · Authors · 2017-12-08
**A rebutted paper version**


We have uploaded a revised version of our paper where we have carefully considered and integrated all the comments of the reviewers.


In summary, we introduce the following clarifications and modifications that improve the presentation:

a) The abstract is made more appealing and consistent.

b) The motivation for the use of the non-linear transform was clarified by adding additional explanations as suggested (please, see the paragraph 1 in subsection 1.2). Additionally, the table about the most used notations was removed.

c) The introduction of section 2 was changed and simplified to an overview about the proposed concept. The type of the used non-linear transforms and the general modeling concept from introduction was moved to a separate subsection 2.1 The parametric non-linear transform modeling

e) Subsection 2.1 is changed to subsection 2.2 and is divided into 2 subsections one for the learning model and one for the testing model
(an update w.r.t. the old version is that one more paragraph is added to explain the relation between the non-linear transform model used in the learning and
the sparsifying transform model used at test time as well as the main reason behind this particular use of the two different models).

f) A minor modification w.r.t. the clarity of presentation was added to subsection 2.3 (in the initial version subsection 2.2).

g) The section "Sensitivity analysis" was moved to the Appendix to have a more uniform structure and enhance the readability. Subsection 3.2 is moved and placed as subsection 2.4. The typos are corrected and the comments of the reviewers were integrated with additional clarifications whenever possible.


We would like to thank all the reviewers for their considerations, constructive attitude and raised comments that lead to the qualitative improvement of our manuscript.

---

### Author Response · Authors · 2018-01-15
**To all the reviewers**

To all reviewers, we would like to extend the appreciation for taking the necessary time, involvement and effort in reading our initial and rebutted paper version, express our gratitude for all the taken considerations, raised comments and concerns about all aspects of this paper, contributing towards increasing the quality of the manuscript.

---

### Decision · Program_Chairs · 2018-01-29
**ICLR 2018 Conference Acceptance Decision**

**Decision:**

Reject

**Comment:**

This paper proposes an approach for learning a sparsifying transform via a set of nonlinear transforms at learning time.  The presentation needs a lot of work.  The original paper was 17 pages long and very difficult to understand.  The revised paper is 12 pages long, which is still too long for the content.  The paper needs to better distinguish between the major and minor points.  It is still too difficult to judge the contribution.